# More Science with Less: Evaluation of a 3D-Printed Weather Station

Adam Theisen[1,2], Max Ungar[1], Bryan Sheridan[1], and Bradley G. Illston[3]

[1]Cooperative Institute for Mesoscale Meteorological Studies, University of Oklahoma 120 David L. Boren Blvd., Suite 2100 Norman, OK 73072
[2]Argonne National Laboratory 9700 S Cass Ave, Lemont, IL 60439
[3]Oklahoma Mesonet, University of Oklahoma 120 David L. Boren Blvd., Suite 2900 Norman, OK 73072

**Correspondence:** Adam Theisen (atheisen@anl.gov)

**Abstract.** A weather station built using 3D-printed parts and low-cost sensors, based on plans and guidance provided by the University Corporation for Atmospheric Research 3D-Printed Automatic Weather Station Initiative, was deployed alongside an Oklahoma Mesonet station to compare its performance against standard commercial sensors and determine the longevity and durability of the system. Temperature, relative humidity, atmospheric pressure, wind speed and direction, solar radiation, and
precipitation measurements were collected over an eight-month field deployment in Norman, Oklahoma. Measurements were comparable to the commercial sensors except for wind direction, which proved to be problematic. Longevity and durability of the system varied, as some sensors and 3D-printed components failed during the deployment. Overall, results show that these low-cost sensors are comparable to the more expensive commercial counterparts and could serve as viable alternatives for researchers and educators with limited resources for short-term deployments. Long-term deployments are feasible with proper
maintenance and regular replacement of sensors and 3D-printed components.

## 1 Introduction

Low-cost sensors, coupled with three-dimensional (3D) printing technologies, can provide researchers and educators with the ability to create tools and instrumentation at a fraction of the cost of commercial counterparts. The 3D-Printed Automatic Weather Station (3D-PAWS) Initiative was launched by the University Corporation for Atmospheric Research (UCAR) and
the US National Weather Service International Activities Office, with support from the USAID Office of U.S. Foreign Disaster Assistance, in an effort to fill observational gaps in remote, sparsely observed regions (Kucera and Steinson, 2017). Scientists behind the 3D-PAWS Initiative developed and open-sourced robust plans, documentation, and software for the development of a 3D-printed weather station capable of measuring temperature, humidity, atmospheric pressure, ultraviolet (UV) index, wind speed and direction, and rainfall using low-cost commercially available sensors (Table 1). Similar efforts to develop
low-cost weather stations have emerged for small-scale wind farm site selection (Aponte-Roa et al., 2018) and investigating micro-climate processes (Ham, 2013). In addition to weather stations, efforts are expanding towards the creation of other low-cost sensors that could benefit environmental and atmospheric science applications (Ham et al., 2014; Kennedy, 2019). Most efforts related to weather station development have focused on relatively short time periods for evaluation of the sensors and

3D-printed components, however, the 3D-PAWS Initiative has tested and deployed their systems in a long-term operational manner, with approximately nineteen stations deployed worldwide (Kucera and Steinson, 2017).

Evaluation of these low-cost systems against commercial grade instrumentation is important in proving these technologies and enabling adoption on a wider scale. The 3D-PAWS system has thus far been evaluated against all-in-one (Smallwood and Santarsiero, 2019; Aura et al., 2019) and commercial grade sensors (Kucera and Steinson, 2017). Smallwood and Santarsiero deployed a complete station, but only collected a limited amount of data due to data acquisition issues and an analysis was not performed. Aura et al. compared three years of 3D-PAWS data with an all-in-one weather station and found high correlation for atmospheric pressure, moderate correlation for temperature, relative humidity, and wind direction and relatively low correlation for wind speed. Additionally, Aura et al. concluded that routine maintenance was important for these automated weather systems to ensure data availability and quality. 3D-PAWS developers evaluated their system against high quality sensors over a ten month time-frame in Boulder, Colorado, in an environment where temperatures ranged from approximately -25 °C to 37 °C and at the National Oceanic and Atmospheric Administration (NOAA) testbed facility in Sterling, Virginia (Kucera and Steinson, 2017). Results showed good agreement between sensors, with most sensor uncertainty falling within range of the manufacturer specifications. The exception was relative humidity which had a higher uncertainty of 5.7 % (Table 1). Performance of the UV sensor was not assessed in previous comparison studies.

Accurate reference sensors are essential for proper comparisons. Ideally, reference sensors would conform to standards defined in World Meteorological Organization (WMO) Guide to Instruments and Methods of Observation (WMO, 2018). However, the WMO guide also indicates that operational uncertainty conforming to these requirements will not be met in many instances and are only achievable with the "highest quality sensors and procedures". There are a number of organizations that deploy high quality sensors and implement best practices as they relate to calibration and data quality that can serve as viable reference stations. The Oklahoma Mesonet (Mcpherson et al., 2007; Brock et al., 1995), hereafter referred to as Mesonet, deploys high quality meteorological instrumentation (Table 2) in every county across Oklahoma. Mesonet sensors undergo routine maintenance and are rotated out of the field on a regular schedule. Calibrations are performed before and after deployment to the field, leading to well characterized systems (Mcpherson et al., 2007). The accuracy of the Mesonet sensors is comparable to Kucera and Steinson, with noticeable improvements over the all-in-one sensors used in the Smallwood and Santarsiero and Aura et al. studies (Table 3). Information about the particular sensors used in each of these studies is documented in Appendix A.

This study was supported by a grant through the Cooperative Institute for Mesoscale Meteorological Studies (CIMMS) at the University of Oklahoma with the goals of verifying the results from previous inter-comparison studies, assessing the longevity of the sensors and 3D-printed components, and most importantly, providing undergraduate meteorology students with skills they would not otherwise have learned in the classroom. Through this project, students were able to gain valuable hands on experience with proposal writing, project plan development, 3D printing, instrument engineering and development, and field campaign operations.

## 2 Station Configuration

The weather station was built based on specifications provided by the 3D-PAWS Initiative with some modifications. Over 100 parts were 3D-printed using off-white acrylonitrile styrene acrylate (ASA), which has higher ultraviolet radiation, temperature, and impact resistance than regular polylactic acid (PLA) filament. Parts were printed with a grid infill to reduce printing time. ASA is printed using higher temperatures than standard PLA filament, which can lead to warping of prints, as was the case for the radiation shield leafs. Initial prints of the rain gauge funnel using the original design proved problematic with the printer used. The wall thickness of the funnel was increased to resolve the print issues. Due to a vendor shortage of the original off-white ASA, the funnel was printed with gray ASA. It was coated with polyurethane to seal any remaining imperfections in the print. Lab calibrations of the rain gauge were performed and it was adjusted to ensure that each tip routinely held 0.2 mm of water compared to 0.254 mm for the Mesonet rain gauge. The rain gauge screen was created using parts from a failed funnel print. Mosquito netting was zip tied to the ring and placed securely inside the funnel. Plans provided by the 3D-PAWS Initiative called for opaque plastic (PTFE) to shield the UV sensor. In order to reduce costs, an opaque plastic tray from a frozen meal was used to create the UV sensor cover (Fig. 1). Temperature, relative humidity, atmospheric pressure, and UV sensors were all sealed with conformal coating to protect against degradation due to moisture.

Due to limitations with anchoring the station in ground, a frame had to be developed to withstand weather conditions in Oklahoma. The frame was built from standard polyvinyl chloride pipe (PVC) and consisted of a central trunk connected to three legs (Fig. 2). Each leg was connected to a height adjustable concrete footing. In order to minimize vibrations on the tipping bucket rain gauge, the support legs were also set in concrete. In lieu of building a Raspberry Pi tube, an electrical junction box was used to house the Raspberry Pi. Temperature, relative humidity, and atmospheric pressure sensors were installed in the naturally aspirated radiation shield at 1.5 meters to match the height of the Mesonet. Wind direction, wind speed, and UV light sensors were installed on the crossbar at approximately two meters compared to ten meters for the Mesonet wind measurements. The tipping bucket rain gauge was installed at roughly 0.3 meters compared to 0.6 meters for the Mesonet. A secondary temperature sensor was installed in the Raspberry Pi box to monitor internal temperatures.

Software provided by the 3D-PAWS Initiative was not compatible with the Raspberry Pi version used for this study and additional software engineering was required. Existing python libraries were used for communications with temperature (DiCola, 2014b), relative humidity (Gaggero, 2015), atmospheric pressure (DiCola, 2014a), and UV light sensors (Gutting, 2014). The 3D-PAWS software image was decoded and used as a basis for the wind and rain programs. Data were collected instantaneously every minute for temperature, pressure, relative humidity, and UV variables. The rain program was constantly listening for tip events and recorded event totals every minute. The wind program was constantly running as well, taking measurements every ten seconds and recording average, minimum, and maximum wind speed and direction every minute. Programs were set up to automatically start up on any reboot or power loss event to ensure robust operations. Data were automatically uploaded via WiFi connection to a cloud based storage location at the end of every day (24:00:00 UTC) to ensure minimal data loss in the event of a catastrophic failure.

The 3D-printed station was deployed approximately seventy meters to the West-Northwest of the Norman Oklahoma Mesonet station from 15 August 2018 to 15 April 2019 (Fig. 3). The Norman station (Fig. 4) served as an ideal reference point due to the proximity to the University of Oklahoma for easy installation and routine maintenance visits. Power was also easily accessible, eliminating the need for solar panels and batteries. The 3D-printed station cross-arm was oriented perpendicular to North with the rain gauge positioned to the West of the station in order to minimize any interference from the station itself or the larger ten meter tower nearby. The terrain of the site was slightly sloped such that the 3D-printed station was

approximately two meters higher than the Mesonet station. Surrounding vegetation was mostly native grasses, which were mowed on a regular basis.

## 3 Results

Temperature, relative humidity, atmospheric pressure, wind speed and direction, and UV data collected from the 3D-printed

station were averaged to five minutes in order to compare with the Mesonet data downloaded from Atmospheric Radiation Measurement User Facility (ARM, 2019). The Atmospheric data Community Toolkit (Theisen et al., 2020) was used to read in the different data formats into common xarray objects for analysis (Hoyer and Hamman, 2017). The subsequent scatter plots, produced using Matplotlib (Hunter, 2007), follow the same format, the 3D-printed station is displayed on the x-axis and Mesonet on the y-axis. Data points are color-coded by a consistent time interval corresponding to the full length of the

deployment, 15 August 2018 to 15 April 2019, with dark blue indicating data collected towards the beginning of the deployment and yellow indicating data collected towards the end of it. A one-to-one line is indicated by the blue dashed line. The solid black line denotes the linear regression calculated using SciPy (Virtanen et al., 2020). Slope, intercept, and correlation coefficient are listed on the bottom left. Standard error of the mean (SEM), root mean square error (RMSE), average difference, and minimum and maximum values of the 3D-printed station and Mesonet are listed in the lower right. A summary of the RMSE

and correlation coefficients broken down by month and for the full deployment is in Table 4.

### 3.1 Air Temperature

Air temperature data were quality controlled by applying an upper threshold of 45 °C to the 3D-printed station temperature data in order to remove erroneous data points. The Mesonet data had been properly quality controlled and no further quality control was necessary. The MCP9808 sensor reported large values towards the end of the deployment, but otherwise performed well

with a RMSE of 1.22 °C when compared with the Mesonet sensor (Fig. 5). The lowest RMSE value (0.42 °C) was recorded in month seven immediately before the sensor began to fail, which resulted in an RMSE of 1.53 °C for month eight (Table 4). Temperature sensors were also incorporated into the HTU21D relative humidity and BMP280 atmospheric pressure sensors. Temperature from the BMP280 sensor was not included in the analysis due to the sensors subsequent relocation inside the Raspberry Pi box. Temperature reported by the HTU21D sensor performed better than the primary MCP9808 sensor, with an

overall RMSE of 0.97 °C (Fig. 6). However, the HTU21D sensor failed in the sixth month of the deployment. This failure was

attributed to corrosion on the board that was not observed amongst the other sensors. The MCP9808 sensor did show some signs of degradation at the end of the deployment due to moisture but was otherwise in relatively good shape.

Differences in the radiation shield configuration between stations contributed to a portion of the observed differences. The Mesonet deploys actively aspirated radiation shields, while the 3D-printed station radiation shield was naturally aspirated. Data were additionally analyzed by applying thresholds to the data based on wind speeds from 1 m s$^{-1}$ to 8 m s$^{-1}$. RMSE significantly improved for the MCP9808 sensor, from 1.22 °C to 1.08 °C with a threshold of 1 m s$^{-1}$ and from 0.97 °C to 0.91 °C for the HTU21D sensor. RMSE continued to decline with increasing wind speeds for both sensors (Table 5). As flow through the naturally aspirated radiation shields increased, it became comparable to the flow through the Mesonet aspirated radiation shields. The RMSE of both sensors is inline with the sensor uncertainties between both stations (MCP9808 0.8 °C; HTU21D 0.6 °C) when the wind speeds are greater than roughly 5 m s$^{-1}$.

## 3.2   Relative Humidity

As mentioned in the previous section, the HTU21D sensor failed in month six of the deployment, but was able to measure a broad range of relative humidity values from 11 % to 100 % (Fig. 7). Corrections were applied using a manufacturer supplied temperature coefficient compensation equation with a temperature coefficient of -0.15 % RH/°C (Inc, 2013). Values above 100 % were set to 100 %, following the Mesonet practice (Mesonet, b). RMSE for the entire campaign was 3.33 %, indicating a slight moist bias with the low-cost sensor. Prior to applying the manufacturer correction, the overall RMSE was 5.00 %. HTU21D sensor specifications indicate that uncertainties are larger for relative humidity measurements greater than 80% (Table 1). However, the RMSE was little changed (0.09 %) when data above and below this limit were excluded from the analysis. When this limit was lowered to 50 %, relative humidity values over 50 % had an RMSE of 3.44 % compared to 2.42 % under 50 %. Unlike the RMSE of temperature, the relative humidity RMSE was relatively constant with increasing wind speed thresholds (Table 5). A polytetrafluoroethylene (PTFE) filter covered the sensor to keep it clean and appeared to have some staining when the sensor was uninstalled from the station. The filter itself was hydrophobic, but it is possible that the accumulated dust on it was not. Overall, the observed errors between the systems were comparable to the sensor accuracy across relative humidity values.

## 3.3   Atmospheric Pressure

The initial BMP280 pressure sensor deployed with the station had large errors when compared to the Mesonet. A replacement sensor was installed but suffered from communication problems owing to bad wire connections and was moved from the radiation shield to the Raspberry Pi box. The assumption was that there would be minimal difference in pressure measurements due to the openness of the PVC frame and that the connection to the radiation shield would allow for proper air flow for atmospheric pressure measurements. Temperature in the box varied but did not appear to greatly affect the measurements, as the RMSE of the pressure was fairly constant for the entire deployment with an overall RMSE of 2.39 hPa (Table 4). There was very little deviation in the measurements and it appears that the BMP280 sensor has a nearly constant offset when compared to the Mesonet (Fig. 8).

## 3.4 Wind Speed

In order to properly compare the ten meter Mesonet wind speeds to the 3D-printed station two meter winds, a logarithmic wind profile was assumed and the ten meter Mesonet winds were adjusted based on the method from Allen et al. (1998). The resulting conversion factor, 0.748, was applied to the Mesonet wind speed data. Performance over the first three months was comparable to the Mesonet station with an RMSE between 0.56 m s$^{-1}$ and 0.59 m s$^{-1}$ (Table 4). RMSE slowly increased over the course of the deployment which could be attributed to accumulation of dust on the bearing as there was not a procedure in place to routinely clean or oil it. There was a substantial increased in month seven to 2.01 m s$^{-1}$, evident in Figure 9, as the data showed more of an exponential relationship towards the end of the deployment (yellow). The anemometer head started to fail on 16 February 2019, resulting in intermittent measurements of 0 m s$^{-1}$ and significantly impacted wind speed observations. The anemometer failed on 30 March 2019 when the head completely sheared off from the driveshaft, resulting in measurements of 0 m s$^{-1}$ for the remainder of the deployment. Measurements of 0 m s$^{-1}$ from the 3D-printed anemometer were excluded from the analysis between 16 February 2019 to 15 April 2019. This failure could be attributed to two factors. The first factor being the reduced infill used to print the parts in order to save time, which would have weakened the overall strength of the components. The second factor is how the anemometer was built. Initially the anemometer had a large amount of wobble. This wobble was greatly reduced before deployment but not completely eliminated and could have further added to the strain on the driveshaft.

## 3.5 Wind Direction

Throughout the deployment, the wind vane tended to stick in certain directions, most notably around 0 degrees (Fig. 10). Efforts were taken to reduce the sticking by applying lubricant, but it proved to be an issue for the extent of the deployment. In order to reduce the impact of this known problem on the analysis, wind directions within 1 degree of North were excluded from the analysis for both stations. Additionally, the period during which the anemometer was problematic, as previously mentioned, was excluded from the wind rose plots for both stations (Fig. 10). The 3D-printed wind vane was held in place by a 3D-printed clamp and bolt that routinely loosened over time. This caused the orientation to drift throughout the entire deployment which resulted in varying offsets in the data. The alignment was checked and adjusted with each maintenance visit but proved to only be a temporary fix. This drift is noticeable in Figure 10 when comparing the dominant wind direction. Disregarding the erroneous northerly spike in the 3D-printed station data, there is a roughly 10 degree offset in the dominant wind direction. While there was agreement in wind directions at times, the RMSE was large and the correlation coefficient was relatively low for a majority of the campaign (Table 4). In addition to the data quality issues already noted, small grooves in the wind vane, a byproduct of the 3D-printing process, created an ideal location for insects to lay large numbers of eggs which could have impacted measurements in the latter months of the deployment.

### 3.6 Solar Radiation

The Mesonet measured downwelling global solar radiation and the UV sensor measured counts of visible and infrared light in order to calculate a UV index. It was discovered that similar UV sensors from other manufacturers provided coefficients for calculating lux from their sensors, which could then be converted to W m$^{-2}$. However, no such coefficients were found for the SI1145 sensor. There was a linear correlation between the downwelling global measurements and the visible counts (Fig. 11), so a simple linear regression was performed to determine the slope (0.70) and intercept (170.66) of the data for the entire deployment. These values were applied to the counts data to make it comparable with the Mesonet measurements. The SI1145 UV sensor had a high bias for the initial few months of the deployment and a low bias for the latter months. The sensor was difficult to keep perfectly level and was routinely adjusted during maintenance visits. The plastic disc held up to the elements but the glue used to seal it yellowed over time (Fig. 1). 3D-printed connectors routinely lost physical connection to the UV sensor resulting in intermittent outages throughout the deployment. The levelness of the sensors, yellowing of the glue, and intermittent outages contributed to the changes observed in RMSE throughout the deployment (Table 4).

### 3.7 Precipitation

Efforts to increase the sturdiness of the rain gauge funnel failed, as the funnel broke off at the neck on the initial installation. A thick layer of silicone caulk was applied to the break while ensuring that the opening remained clear. The funnel was planned to be replaced upon failure, however, that failure did not occur. The 3D-printed rain gauge performed surprisingly well early on in the deployment compared to the Mesonet (Fig. 12 and 13). Daily accumulations and rain rates compared well with the Mesonet for the first few months (Table 4). The 3D-printed rain gauge performed very well during a heavy precipitation event in month five, measuring a maximum rain rate of 103.2 mm hr$^{-1}$ compared with the Mesonet maximum rain rate of 103.7 mm hr$^{-1}$ (Table 6). Very poor correlation coefficient and low RMSE in month six can be attributed to the limited precipitation recorded for that period of time, with a maximum accumulation of 2.2 mm during that period. The rain gauge tended towards a high bias in months seven and eight, the cause of which is unknown but assumed to be related to its subsequent failure. The nut holding the rain gauge loosened towards the end of the deployment and eventually the wiring disconnected resulting in a number of missed events. Neither the Mesonet nor the 3D-printed rain gauge were heated, so the differences in precipitation measured during solid precipitation events could be attributed to different melt rates between the gauges. The rain gauge was printed with gray filament due to limited supplies of the white ASA filament and was further exposed to the environment than the Mesonet gauge, both of which could contribute to different melt rates.

## 4 Conclusions

While the sensors used as a reference were not up to WMO standards, they were very well maintained and characterized, leading to high confidence in the reference measurements and results. Though there were a number of differences in the physical deployment of the systems, the temperature, relative humidity, atmospheric pressure, wind speed, and precipitation

sensors all performed reasonably well for a majority of the campaign with relatively lower RMSE and higher correlation coefficients observed in comparisons with the Mesonet system. Decreasing RMSE for temperature with increasing wind speeds reveals the effect of the different types of radiation shields. The addition of a miniature five volt fan to the radiation shield to increase airflow could improve the overall temperature and relative humidity measurements, but would add additional strain to the system if operating remotely on solar and battery power. Wind measurements were taken at two different heights, with

the Mesonet wind speed adjusted using an assumption that the wind profile was logarithmic. This assumption may not hold in more complicated terrain where these systems are sometimes deployed. Additional work is need to determine the feasibility of deploying wind sensors at ten meters as it would put added stress on the frame and potentially increase the risk of failure to the entire system. Disappointing results from the wind vane can be attributed to the bearing, but also indicates that a more robust solution is needed to ensure the sensor stays oriented with true North. While comparisons from the solar radiation sensors were

generally positive, it was not a one-to-one comparison and a definitive conclusion cannot be derived. Depending on the needs of the project, different lux or even spectral sensors could be deployed in place of the UV sensor. Additionally, a convex lens could be utilized as an alternative to the flat top in order to provide better measurements off-zenith.

As previously mentioned, some of the 3D-printed components failed (anemometer, rain gauge funnel) or routinely disconnected (UV sensor) but overall the components and the frame held up well to the environmental stresses. The decision to reduce

the print quality by decreasing the infill did have a negative impact on some of the components but in general, the majority performed as expected. Water intruded into the PVC cross-arm through the physical connectors between the 3D-printed parts and the PVC frame. Applying silicone to these areas and drilling holes in the PVC frame alleviated water intrusion and accumulation. Holes were initially drilled into the elbow leading into the Raspberry Pi box to prevent water intrusion and worked as expected. In order to account for the additional temperature sensor in the Raspberry Pi box and the eventual relocation of

the pressure sensor there as well, block connectors were utilized to simplify connections. These block connectors could easily replace the 3D-printed common rail assemblies in order to reduce the assembly time and ensure more reliable connections. The frame and sensor housings that made it to the end of the deployment were donated to the CIMMS education and outreach group.

Overall, the results are positive and indicate that many of these low-cost sensors and the 3D-printed housings can be viable

options for gathering meteorological data for short-term deployments (approximately 6 months) when the cost of commercial sensors is prohibitive. Long-term deployments would require routine maintenance and replacement of the sensors and 3D-printed components to ensure accurate readings and avoid failures. Comparative studies, such as this, will improve the understanding of how well these low-cost sensors can perform, their longevity in the field, and the long-term resource requirements and maintenance schedules for these types of systems. Capabilities in the area of low-cost sensors are constantly

expanding, as are the possibilities for new advancements with other measurements. Subsurface, spectral solar radiation, and aerosol measurements are examples of areas that could benefit from the broader use of low-cost sensors. However, in order to enable wider adoption of these technologies, they must be vetted by the community to ensure that the measurements they provide are comparable to that of industry standard sensors.

*Code and data availability.* Code and data used on the 3D-printed weather station and for the subsequent analysis are available at https: //github.com/AdamTheisen/3DWxSt (Theisen, 2019)

## Appendix A:  Reference Instrumentation

The type and configuration of the sensors used as a reference for the 3D-PAWS comparison studies have varied. This appendix is to document the reference sensors of previous studies and the configurations of those systems if known.

### A1    Smallwood and Santarsiero

Smallwood and Santarsiero used an AcuRite Pro Model 01024 all-in-one weather stations as a reference station. Documentation indicates that barometric pressure is measured but does not list the accuracy of the measurement (AcuRite, 2019). The sensor uses a cup and vane to measure wind speed and direction but is also only capable of measuring sixteen points of wind direction which is why accuracy was left out of Table 3.

### A2    Aura et al.

Aura et al. used an ATMOS41 deployed as part of the Trans-African Hydro-Meteorological Observatory as a reference station. The system does not have moving parts so the wind measurements are from acoustic sensors at two meters. Likewise, the rain gauge sensor uses a drip counter made of gold electrodes (METER Group Inc, 2017a, b).

### A3    Kucera and Steinson NCAR Testbed

The NCAR Marshall Field Site used in Kucera and Steinson housed a variety of sensors. Temperature and humidity were measured with a Campbell Scientific 500 Series Sensor. The NCAR field site website points to a Campbell Scientific HC2S3-L probe being used and the accuracy specifications from that sensor were used (Scientific, 2020). Atmospheric pressure was measured using a Vaisala PTB101B (Scientific, 2017). Wind speed and direction were measured using an R. M. Young propeller and vane (Company, 2020). The precipitation reference sensor was a Geonor T-200B weighing bucket rain gauge (Geonor, 2010).

### A4    Kucera and Steinson NOAA Testbed

The sensors used as reference sensors at the NOAA testbed of the Kucera and Steinson study were slightly different from the NCAR Marshall Field Site. Temperature and humidity were measured using a Technical Services Laboratory, Inc Hygrother-mometer model 1088 (Laboratories, 2018). The vendor information provided accuracy results for the temperature, but the accuracy of the relative humidity measurements was not given. It is unclear if the accuracy of the system given for temperature is the same accuracy of the dew point temperature measurements. The uncertainty of the dew point measurements was found in an Atmospheric Radiation Measurement Program METTWR Handbook (Ritsche, 2006). The accuracy of the barometer was difficult to determine and information was found in a General Service Administration (GSA) schedule (Scientific, 2017). Wind

speed and direction were measured using a Vaisala WS524 ultrasonic wind sensor (Vaisala, 2010). Precipitation was measured using the OTT all-weather precipitation accumulation weighing bucket gauge (White et al., 2004).

**A5   Oklahoma Mesonet**

The Oklahoma Mesonet deploys a R. M. Young Model 41342 temperature probe (Company, b) at 1.5 m (Mesonet, c). Relative humidity measurements are taken at 1.5 m (Mesonet, b) using a Vaisala HUMICAP HMP155 probe (Vaisala, 2019). Atmospheric pressure is measured using a Vaisala PTB220 digital barometer (Vaisala, 2005) and is housed in the data logger enclosure (Mesonet, a). Wind measurements are measured at ten meters (Mesonet, d) using a R. M. Young model 05103 pro-285 peller and vane wind monitor (Company, a). Rainfall is measured using a MetOne tipping bucket rain gauge (Mcpherson et al., 2007). A LI-COR pyranometer (Scientific, 1996) is used to measure downwelling global solar radiation (Mesonet, a).

*Author contributions.* Adam Theisen oversaw the general project, testing sensors, troubleshooting while at CIMMS. Final analysis of the data and development of the manuscript was performed by Adam Theisen at Argonne National Laboratory. Max Ungar and Bryan Sheridan were heavily involved throughout the life cycle of the project. They printed the components, developed the frame, built the wiring harnesses, assembled the weather station, performed site checks, and resolved problems while in the field. Bradley Illston provided insight and direction for deploying the instrument at the Mesonet site and general recommendations for the project.

*Competing interests.* The co-PI of the 3D-PAWS Initiative, Paul Kucera, was an advisor to Adam Theisen as an undergraduate student at the University of North Dakota.

*Acknowledgements.* The authors would like to acknowledge the Cooperative Institute for Mesoscale Meteorological Studies (CIMMS) at the University of Oklahoma and their support of this research effort through the Director's Directed Research Fund grant. We would like to thank Jamie Foucher from CIMMS, for handling all of our orders with ease. We would also like to acknowledge Paul Kucera and Martin Steinson at University Corporation for Atmospheric Research who along with the US National Weather Service International Activities Office launched the 3D-PAWS Initiative and open-sourced the plans with support from the USAID Office of U.S. Foreign Disaster Assistance. The authors would like to acknowledge Brandt Smith, Tyler Thibodeau, and the Tom Love Innovation Hub at the University of Oklahoma for their support and use of their 3D printers, lab space, tools, and consumables. The project would not have been as successful without the use of their facilities. Lastly, the authors acknowledge Alan Perry for his assistance in the design of the concrete footings to allow for the leveling of the frame.

Development of this manuscript was supported by the U.S. Department of Energy, Office of Science, under contract number DE-AC02-06CH11357.

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

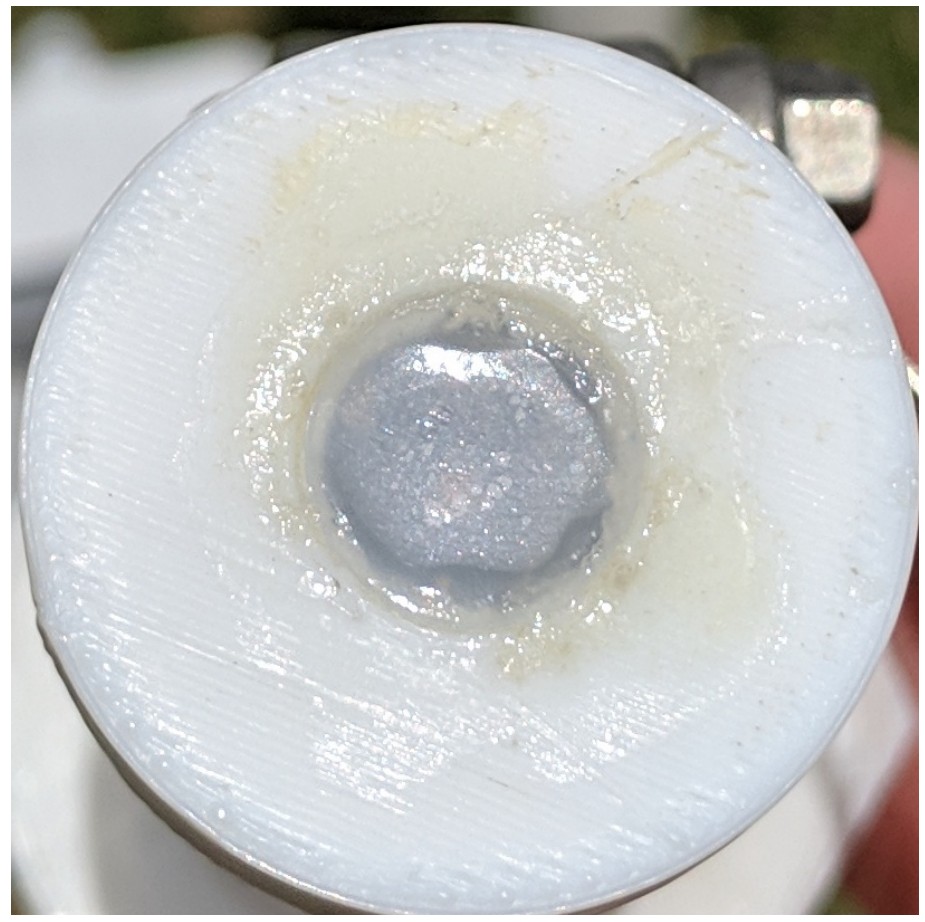

**Figure 1.** Ultraviolet index sensor using a plastic covering cut from a freezer meal tray. Image taken at the end of campaign shows yellowing of the glue used to seal the edges.

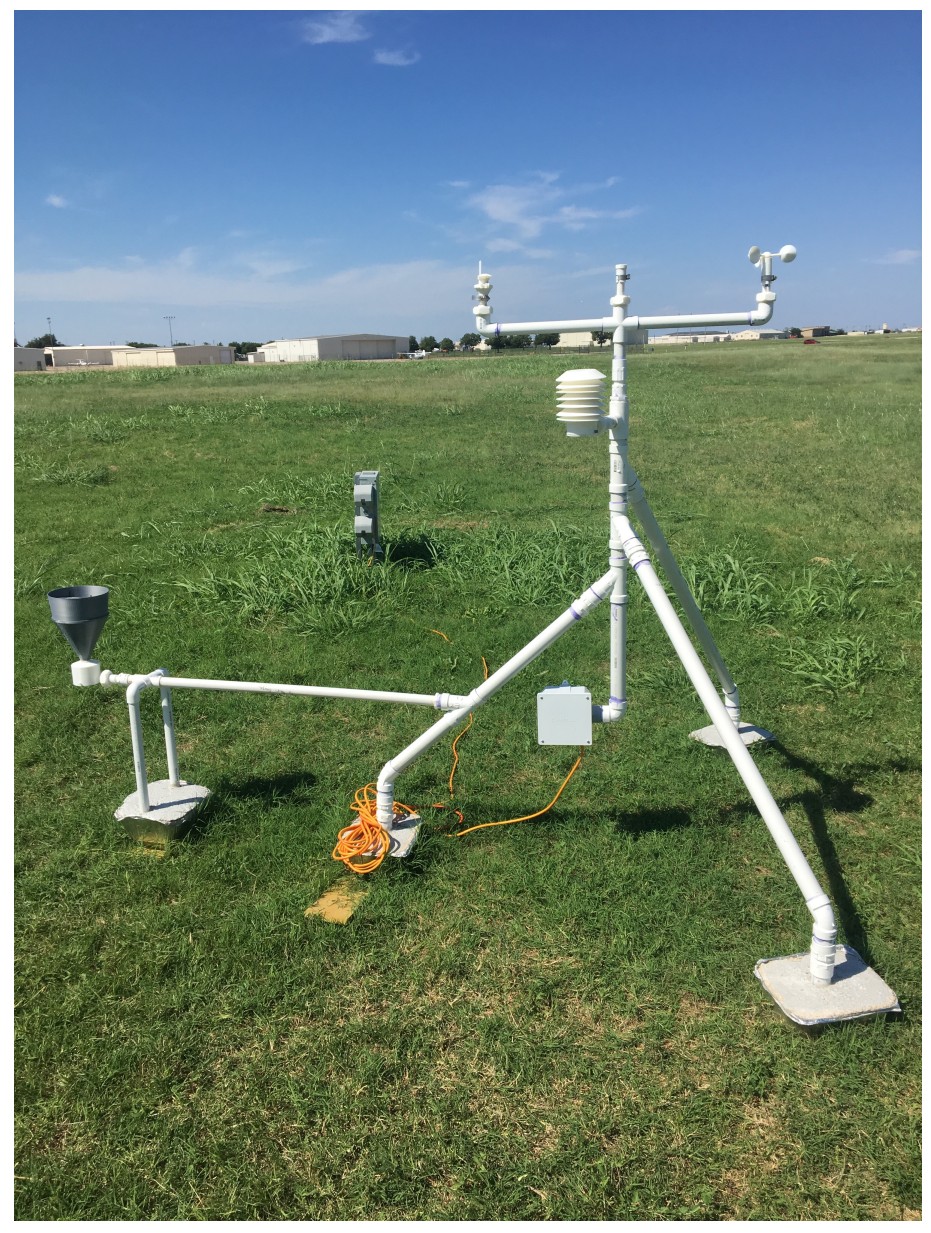

**Figure 2.** 3D-printed weather station upon initial installation in the field.

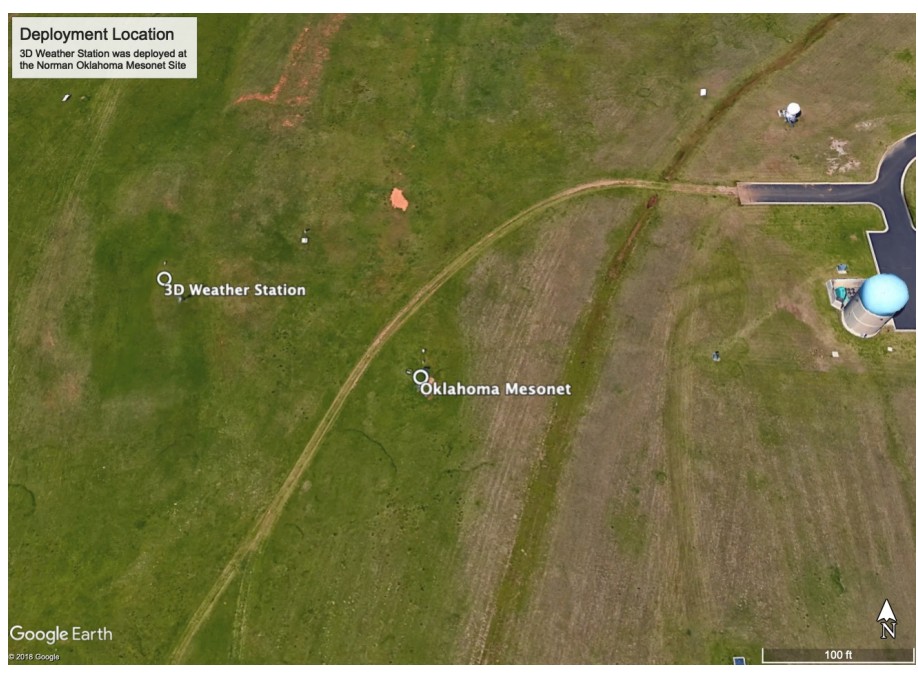

**Figure 3.** Location of the 3D-printed weather station relative to the Oklahoma Mesonet. Image courtesy of Google Earth (Google, 2018)

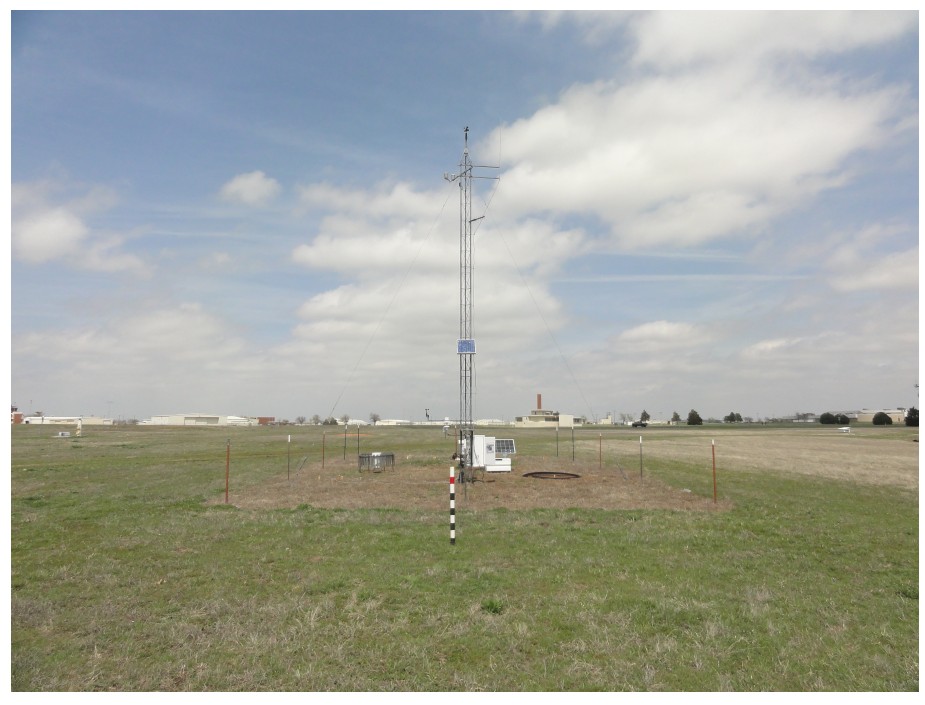

**Figure 4.** Norman Mesonet station from spring 2013. Image courtesy of the Oklahoma Mesonet (Mesonet, 2013)

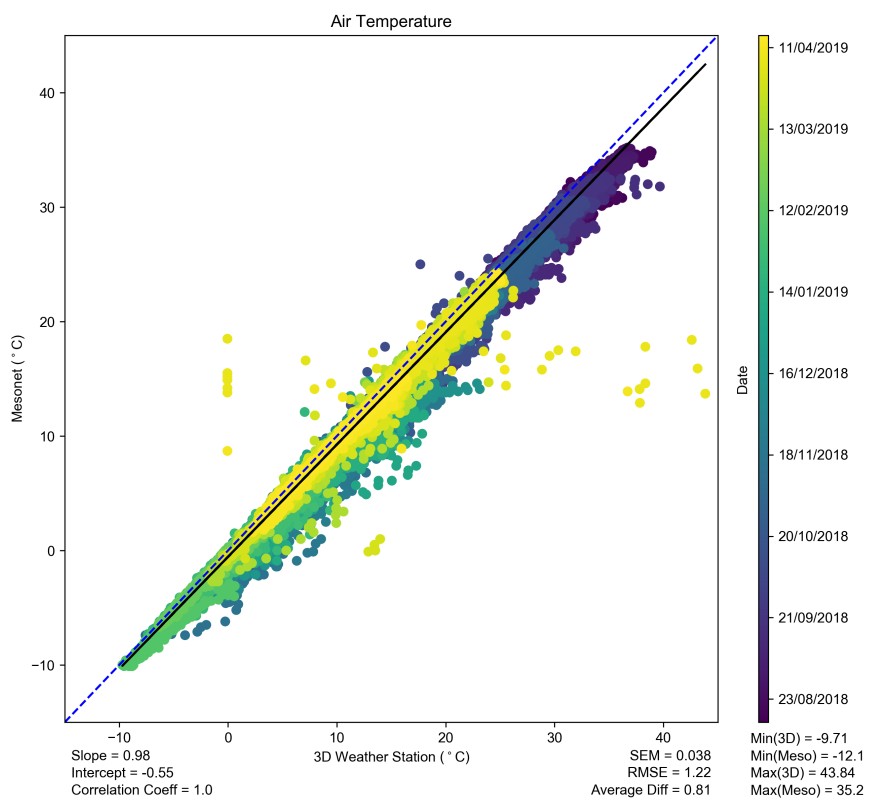

**Figure 5.** Comparison of the low-cost MCP9808 temperature sensor (x) and the Oklahoma Mesonet (y) for the entire deployment, color-coded by time. One-to-one reference line (blue dashed) and linear regression line (black solid) are overlaid.

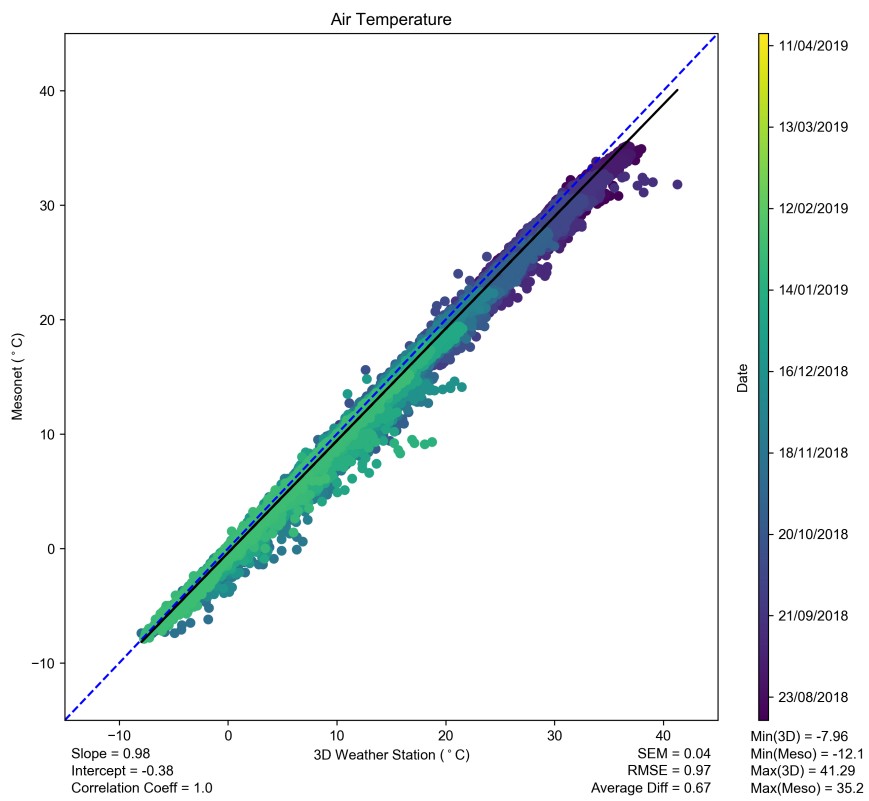

**Figure 6.** Comparison of the temperature from the low-cost HTU21D sensor (x) and the Oklahoma Mesonet (y) for six months of the deployment, color-coded by time. One-to-one reference line (blue dashed) and linear regression line (black solid) are overlaid.

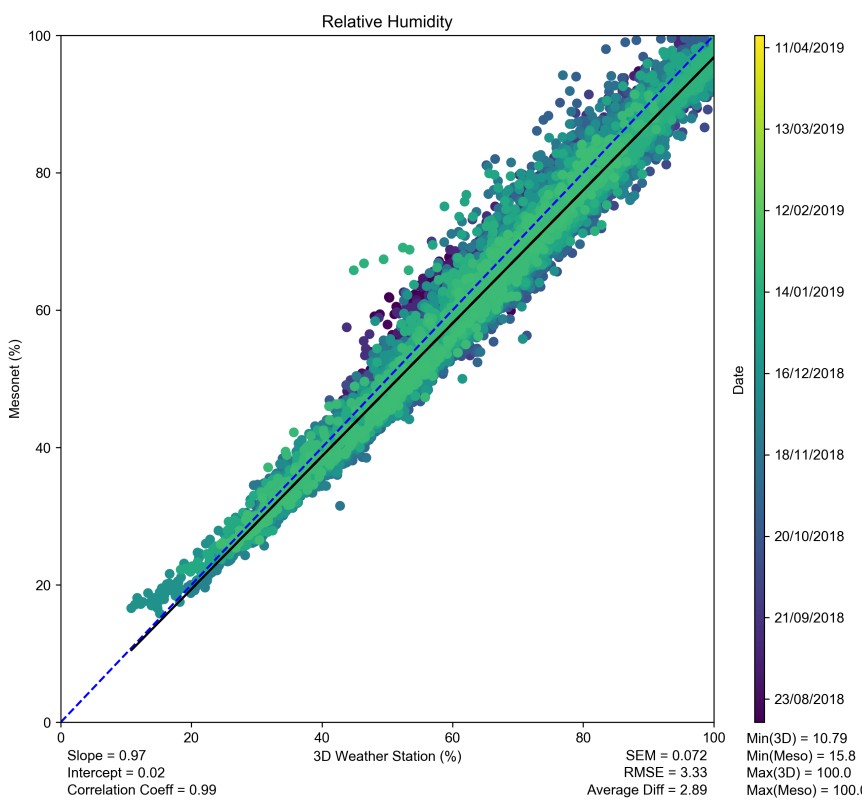

**Figure 7.** Comparison of the low-cost HTU21D relative humidity sensor (x) and the Oklahoma Mesonet (y) for six months of the deployment, color-coded by time. One-to-one reference line (blue dashed) and linear regression line (black solid) are overlaid.

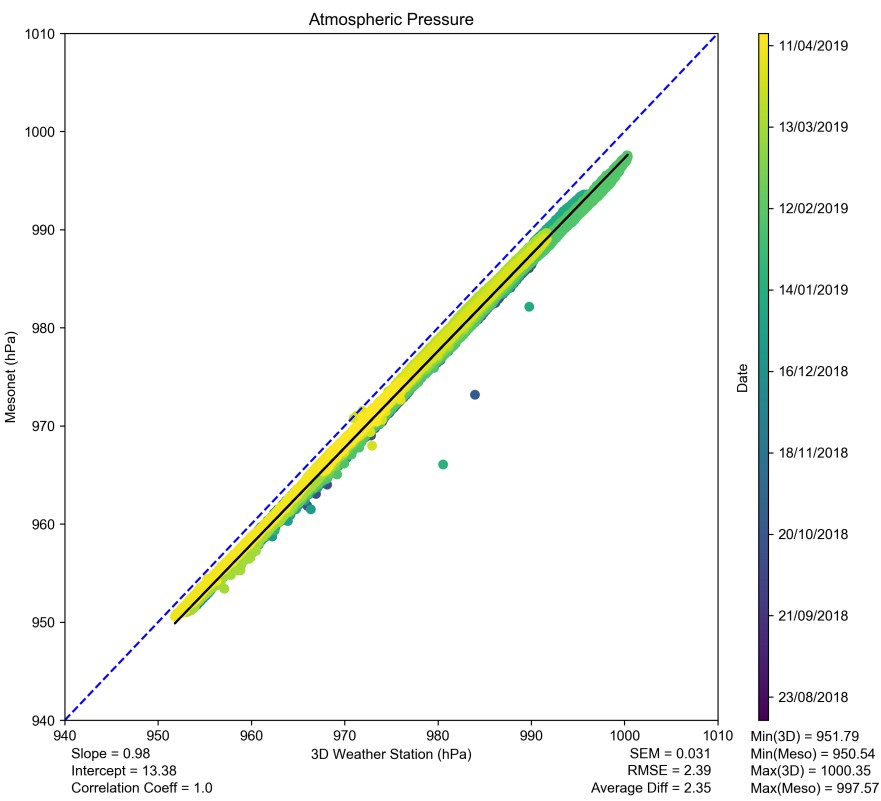

**Figure 8.** Comparison of the low-cost BMP280 pressure sensor (x) and the Oklahoma Mesonet (y) for seven months of the deployment, color-coded by time. One-to-one reference line (blue dashed) and linear regression line (black solid) are overlaid.

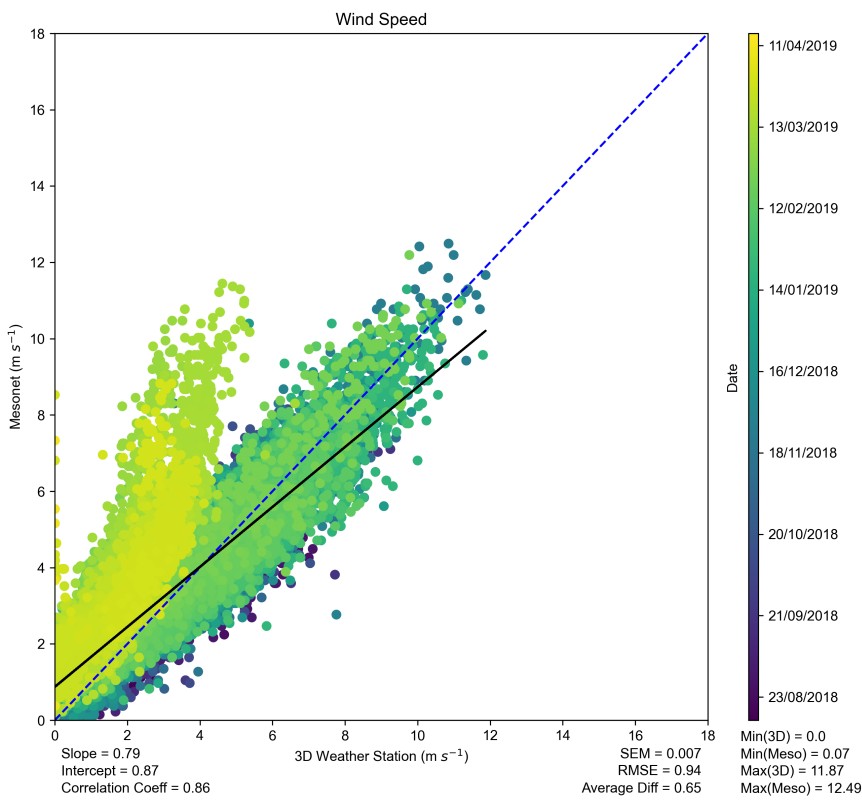

**Figure 9.** Comparison of the 3D-printed anemometer using a Hall effect sensor (x) and the Oklahoma Mesonet (y) for the entire deployment, color-coded by time. One-to-one reference line (blue dashed) and linear regression line (black solid) are overlaid.

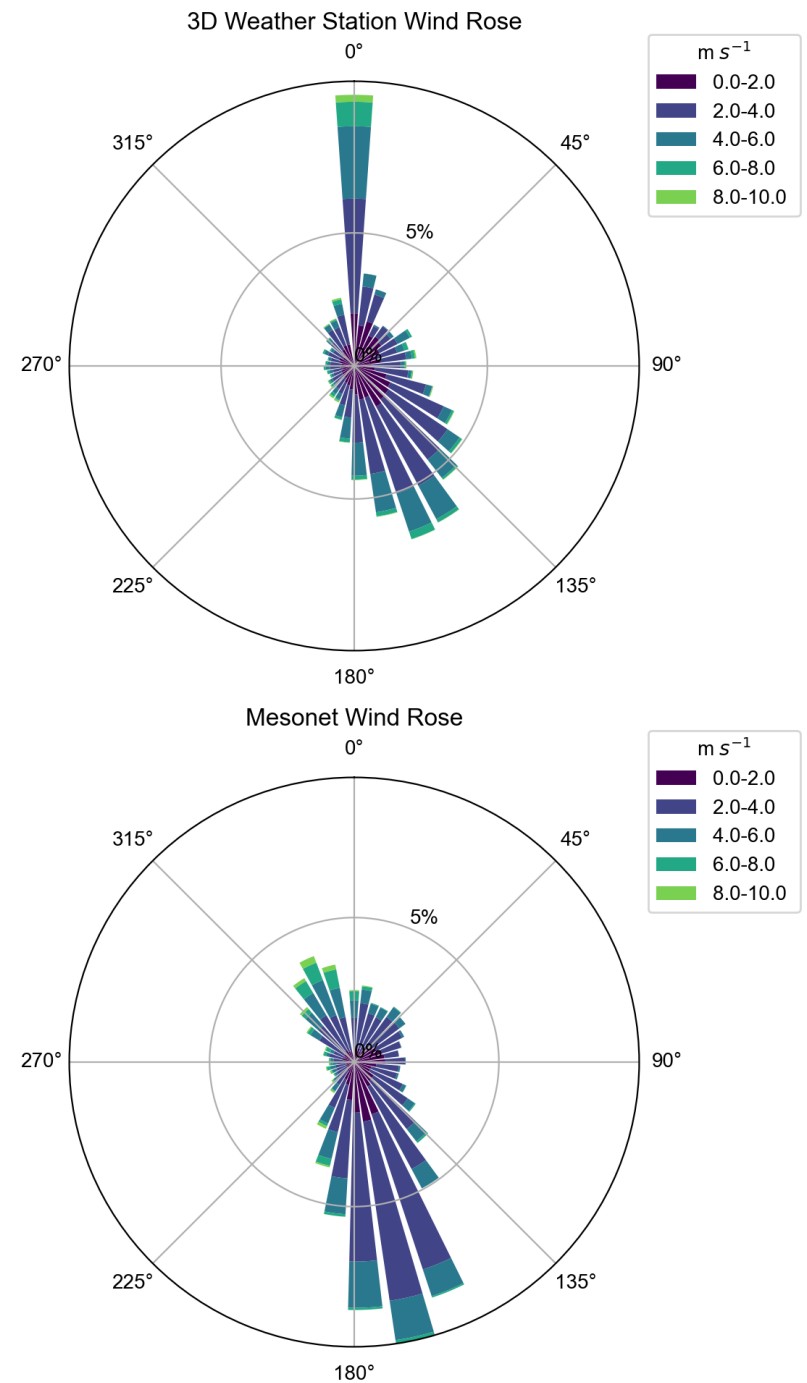

**Figure 10.** Wind roses for the 3D-printed wind vane using a Hall effect rotary sensor (top) and the Oklahoma Mesonet (bottom) for the entire deployment. Color coding is based on the percent occurrence of wind speeds noted in the legend.

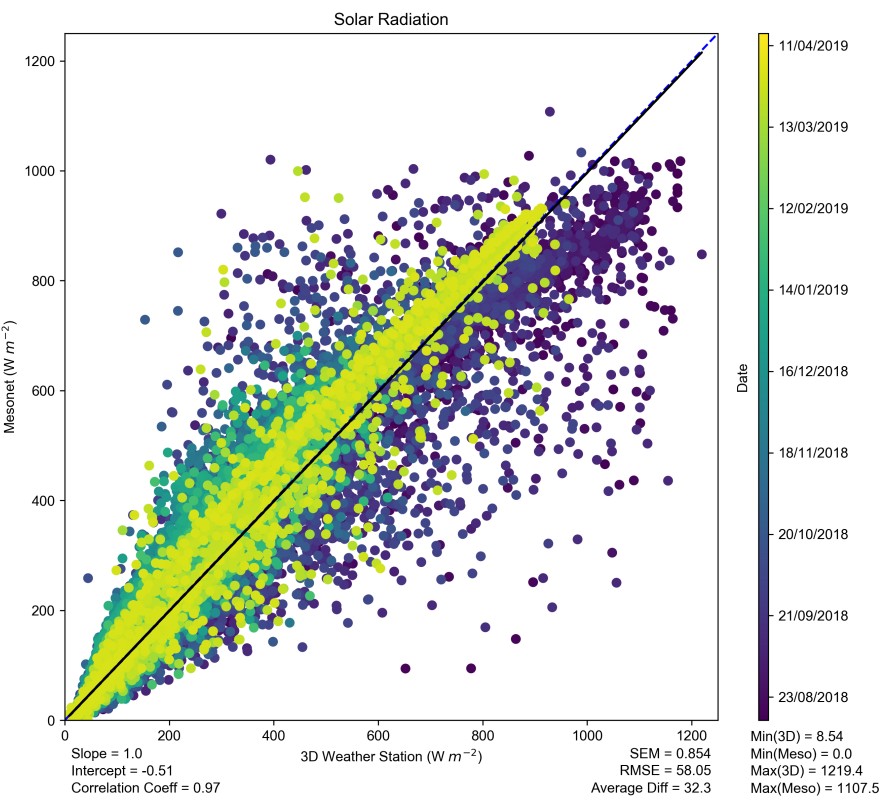

**Figure 11.** Comparison of the low-cost SI1145 UV sensor (x) and the Oklahoma Mesonet downwelling global solar radiation (y) for the entire deployment, color-coded by time. One-to-one reference line (blue dashed) and linear regression line (black solid) are overlaid.

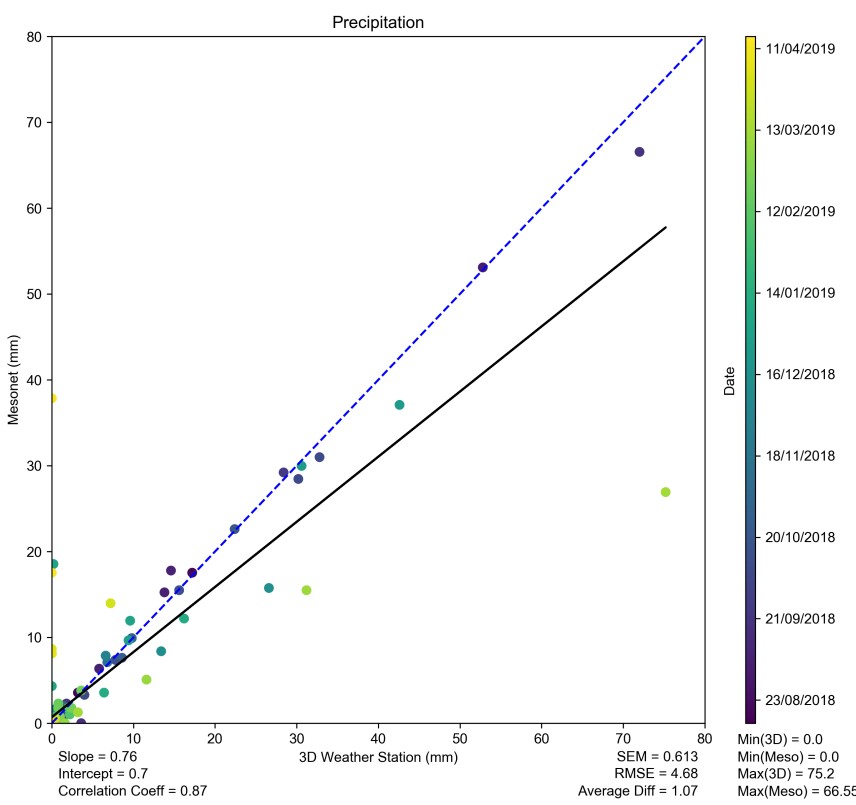

**Figure 12.** Comparison of the daily precipitation accumulations for the 3D-printed tipping bucket rain gauge using a Hall effect sensor (x) and the Oklahoma Mesonet (y) for the entire deployment, color-coded by time. One-to-one reference line (blue dashed) and linear regression line (black solid) are overlaid.

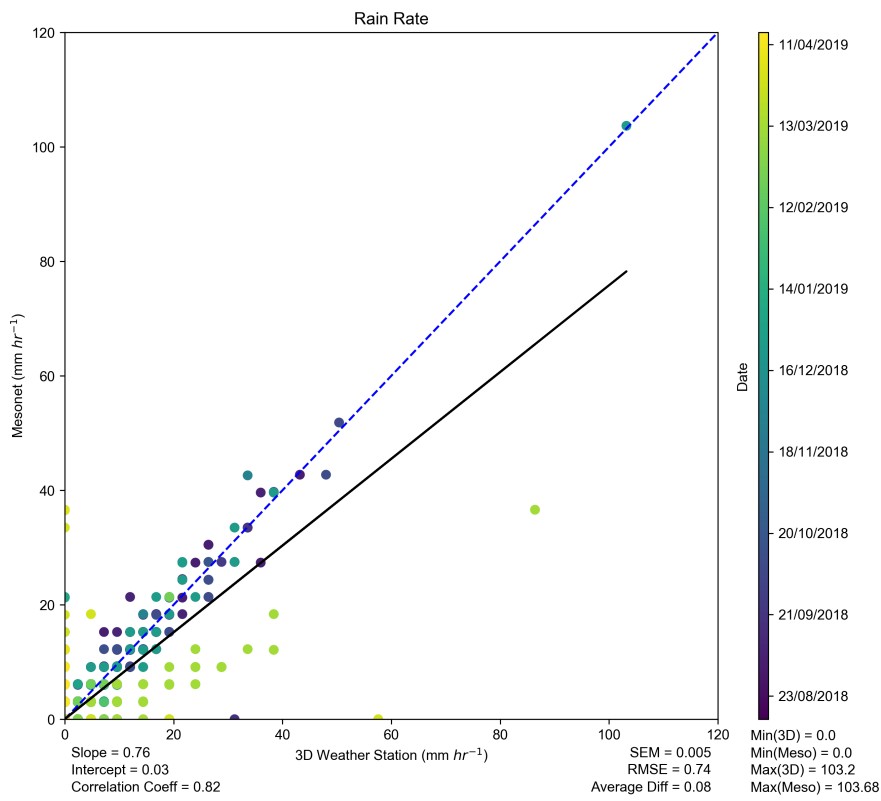

**Figure 13.** Comparison of precipitation rates for the 3D-printed tipping bucket rain gauge using a Hall effect sensor (x) and the Oklahoma Mesonet (y) for the entire deployment, color-coded by time. One-to-one reference line (blue dashed) and linear regression line (black solid) are overlaid.

**Table 1.** 3D weather station sensor specifications

| Parameter | Sensor | Range | Resolution | Accuracy | Kucera & Steinson Uncertainty | Mode | Sampling Time |
|---|---|---|---|---|---|---|---|
| Air Temperature | MCP9808 | -40–125 °C | 0.0625 °C | ± 0.5 °C | ± 0.4 °C | I | 60 s |
| Air Temperature | HTU21D | -40–125 °C | 0.01 °C | ± 0.3 °C (5–60 °C) | NA | I | 60 s |
| Relative Humidity | HTU21D | 0–100 % | ± 0.04 % | ± 3 % (80–100 %) ± 2 % (0–80 %) | ± 5.7 % | I | 60 s |
| Atmospheric Pressure | BMP280 | 300–1100 hPa | 0.16 hPa | ± 1 hPa | ± 0.4 hPa | I | 60 s |
| Wind Speed | SS451A | unknown | 0.1 m s$^{-1}$ | unknown | ± 0.8 m s$^{-1}$ | A | 10 s |
| Wind Direction | Rotary | 0–360 ° | 1 ° | unknown | ± 5 ° | A | 10 s |
| UV Index | SI1145 | unknown | 0.282 ct lux$^{-1}$ | unknown | NA | I | 60 s |
| Rainfall | SS451A |  | 0.2 mm | unknown | 10 % | T |  |

Wind speed resolution from Kucera and Steinson results

Mode definitions: I = Instantaneous measurement; A = Average over time; T = Total

**Table 2.** Oklahoma Mesonet instrumentation (Mcpherson et al., 2007)

| Parameter | Sensor | Range | Resolution | Accuracy | Mode | Sampling Time |
|---|---|---|---|---|---|---|
| Air Temperature | RM Young 41342 RTD Probe | -50–50 $^{\circ}$C (calibrated) | 0.01 $^{\circ}$C | ± 0.3 $^{\circ}$C at 23 $^{\circ}$C | A | 3 s |
| Relative Humidity | Vaisala HMP155 | 0–100 % | 0.03 % | ± 1 % (40–97 %) ± 0.6 % (0–40 %) | A | 3 s |
| Atmospheric Pressure | Vaisala Barometer | 500–1100 hPa | 0.1 hPa | ± 0.2 hPa | A | 12 s |
| Wind Speed | RM Young Wind Monitor | 0–100 m s$^{-1}$ | 0.03 m s$^{-1}$ | ± 1 % or 0.3 m s$^{-1}$ | A | 3 s |
| Wind Direction | RM Young Wind Monitor | 0–360 $^{\circ}$ | 0.05 $^{\circ}$ | ± 3 $^{\circ}$ | A | 3 s |
| Solar Radiation | Li-Cor Pyranometer | 0–3000 W m$^{-2}$ | 0.23 W m$^{-2}$ | ± 5 % | A | 3 s |
| Rainfall | Met One TBRG | | 0.25 mm | 1 % (2.5–7.6 cm hr$^{-1}$) at 21 $^{\circ}$C | T | |

Mode definitions: I = Instantaneous measurement; A = Average over time; T = Total

Temperature accuracy does not include the added uncertainty from the radiation shield

**Table 3.** Summary of reference instrument accuracy used in 3D-PAWS comparison studies, including requirements from WMO Guide to Instrument and Methods of Observation - Volume 1 Annex 1.A. (WMO, 2018)

| Parameter | Smallwood & Santarsiero | Aura et al. | Kucera & Steinson NCAR Testbed | Kucera & Steinson NOAA Testbed | Oklahoma Mesonet | WMO Guidelines* |
|---|---|---|---|---|---|---|
| Air Temperature | 1.1 °C | 0.6 °C | 0.1 °C at 23 °C | 0.28 °C (-50–50 °C) | 0.3 °C at 23 °C | 0.1 °C (-40–40 °C) AMU: 0.2 °C |
| Relative Humidity | 5 % (90–100 %) 4 % (80–90 %) 3 % (20–80 %) 4 % (10–20 %) 5 % (1–10 %) | 4 % (90–100 %) 2 % (15–90 %) | 0.8 % at 23 °C | Dewpoint Temperature 1 °C (-1–30 °C) | ± 1 % (40–97 %) ± 0.6 % (0–40 %) | 1 % AMU: 3 % |
| Atmospheric Pressure | | 1 hPa | 0.5 hPa | 0.1 hPa | 0.2 hPa | 0.1 hPa AMU: 0.15 hPa |
| Wind Speed | Accuracy in m s$^{-1}$ 2.2 (<44 m s$^{-1}$) 1.8 (<22 m s$^{-1}$) 1.3 (<13 m s$^{-1}$) 0.9 (<4.5 m s$^{-1}$) | 3 % | Greater of 0.3 m s$^{-1}$ or 3 % | Greater of 0.135 m s$^{-1}$ or 3 % | Greater of 1 % or 0.3 m s$^{-1}$ | 0.5 m s$^{-1}$ (<5 m s$^{-1}$) 10 % (> 5 m s$^{-1}$) AMU: Not Listed |
| Wind Direction | | 5 ° | 3 ° | 2 ° | 3 ° | 5 ° AMU: 5 ° |
| Solar Radiation | NA | 5 % | NA | NA | 5 % | 2 % AMU: Daily: 5 % AMU: Hourly 8 % |
| Rainfall | 5 % | 5 % | 0.1 % FS | 4 % | 1 % (2.5–7.6 cm hr$^{-1}$) at 21 °C | 0.1 mm (<5mm) 2 % (> 5mm) AMU: Greater of 5 % or 0.1 mm |

AMU: WMO Achievable Measurement Uncertainty

\* - WMO Guide to Instruments and Methods of Observation, Volume 1 - Measurement of Meteorological Variables, Annex 1.A (WMO, 2018)

Information was retrieved from a number of sources, see Appendix A for details

Temperature accuracy does not include the added uncertainty from the radiation shield

**Table 4.** Comparison statistics summary of RMSE (top value) and correlation coefficient (bottom value)

| Parameter | Month 1 15 Aug 2018 | Month 2 16 Sep 2018 | Month 3 16 Oct 2018 | Month 4 16 Nov 2018 | Month 5 16 Dec 2018 | Month 6 16 Jan 2019 | Month 7 16 Feb 2019 | Month 8 16 Mar 2019 | Entire Period |
|---|---|---|---|---|---|---|---|---|---|
| MCP Air Temperature (°C) | 1.32 0.98 | 1.06 0.99 | 1.26 0.99 | 1.33 0.99 | 1.20 0.99 | 0.90 0.99 | 0.42 1.00 | 1.53 0.98 | 1.22 1.00 |
| HTU Air Temperature (°C) | 1.11 0.99 | 0.94 0.99 | 0.96 1.00 | 0.99 0.99 | 0.85 0.99 | 1.00 0.99 | | | 0.97 1.00 |
| Relative Humidity (%) | 2.63 0.99 | 3.38 0.99 | 3.73 0.99 | 3.42 0.99 | 3.45 0.99 | 3.09 0.99 | | | 3.33 0.99 |
| Atmospheric Pressure (hPa) | | 2.92 1.00 | 2.51 1.00 | 2.21 1.00 | 2.44 1.00 | 2.60 1.00 | 2.41 1.00 | 1.90 1.00 | 2.39 1.00 |
| Wind Speed (m s$^{-1}$) | 0.58 0.92 | 0.56 0.94 | 0.59 0.95 | 0.67 0.95 | 0.69 0.94 | 0.88 0.94 | 2.01 0.76 | 1.7 0.82 | 0.94 0.87 |
| Wind Direction (°) | 74.08 0.57 | 62.98 0.71 | 60.73 0.85 | 62.35 0.81 | 57.64 0.83 | 106.63 0.45 | 94.19 0.67 | 102.5 0.45 | 78.28 0.70 |
| Solar Radiation (W m$^{-2}$) | 81.64 0.98 | 68.77 0.96 | 50.86 0.98 | 52.35 0.99 | 44.49 0.98 | 34.14 0.99 | 34.23 0.99 | 54.22 0.99 | 58.05 0.97 |
| Rainfall Daily Total (mm) | 0.65 1.00 | 1.32 1.00 | 0.27 1.00 | 2.21 0.98 | 3.79 0.92 | 0.26 0.89 | 9.87 0.98 | 8.02 0.25 | 4.68 0.87 |
| Rain Rate (mm hr$^{-1}$) | 0.42 0.96 | 0.61 0.94 | 0.35 0.87 | 0.34 0.86 | 0.69 0.91 | 0.20 0.33 | 1.27 0.88 | 1.25 0.01 | 0.74 0.82 |

Date indicated in first row is the start date of the period used in the analysis.

**Table 5.** 3D weather station temperature RMSE (top value) and correlation coefficient (bottom value) response to increased wind speeds thresholds

| Sensor | 0 m s$^{-1}$ | 1 m s$^{-1}$ | 2 m s$^{-1}$ | 3 m s$^{-1}$ | 4 m s$^{-1}$ | 5 m s$^{-1}$ | 6 m s$^{-1}$ | 7 m s$^{-1}$ | 8 m s$^{-1}$ |
|---|---|---|---|---|---|---|---|---|---|
| MCP9808 Temperature (°C) | 1.22 1.00 | 1.08 1.00 | 1.01 1.00 | 0.94 1.00 | 0.88 1.00 | 0.75 1.00 | 0.60 1.00 | 0.54 1.00 | 0.51 1.00 |
| HTU21D Temperature (°C) | 0.97 1.00 | 0.91 1.00 | 0.87 1.00 | 0.83 1.00 | 0.77 1.00 | 0.65 1.00 | 0.50 1.00 | 0.43 1.00 | 0.47 1.00 |
| HTU21D Relative Humidity (%) | 3.33 0.99 | 3.35 0.99 | 3.28 0.99 | 3.27 1.00 | 3.33 1.00 | 3.41 1.00 | 3.56 1.00 | 3.59 1.00 | 3.49 1.00 |

**Table 6.** Maximum precipitation rate and daily accumulations recorded each month

| Parameter | Month 1 15 Aug 2018 | Month 2 16 Sep 2018 | Month 3 16 Oct 2018 | Month 4 16 Nov 2018 | Month 5 16 Dec 2018 | Month 6 16 Jan 2019 | Month 7 16 Feb 2019 | Month 8 16 Mar 2019 |
|---|---|---|---|---|---|---|---|---|
| Mesonet Accumulation (mm) | 53.1 | 66.6 | 22.6 | 15.8 | 37.1 | 1.8 | 26.9 | 37.9 |
| 3D-Printed Accumulation (mm) | 52.8 | 72.0 | 22.4 | 26.6 | 42.6 | 2.2 | 75.2 | 7.2 |
| Mesonet Rain Rate (mm hr$^{-1}$) | 42.7 | 51.8 | 27.5 | 42.6 | 103.7 | 6.1 | 36.6 | 36.6 |
| 3D-Printed Rain Rate (mm hr$^{-1}$) | 43.2 | 50.4 | 26.4 | 33.6 | 103.2 | 7.2 | 86.4 | 57.6 |