# Peer review of "More Science with Less: Evaluation of a 3D-Printed Weather Station"

_Atmospheric Measurement Techniques, 2019_

## Referee Comment (RC1) · Anonymous Referee #3 · 16 Mar 2020

The manuscript presents the evaluation of low-cost meteorological sensors positioned inside a 3D-printed weather station. This evaluation was carried out by positioning the 3D-printed weather station close to a reference weather station belonging to the Oklahoma Mesonet network.

The study lasted eight months. The collected data were compared with the aim of both studying the performance of low-cost sensors compared to standard sensors, and to determine the longevity and resistance of the 3D-printed weather station.

The authors evaluated the performance of low-cost instruments by calculating the average difference and the correlation coefficient respect to data of the reference station.

The results are presented through a series of scatter plots for the various geophysical

quantities (temperature, relative humidity, atmospheric pressure, wind speed and direction, UV radiation and precipitation) in which the data are also color-coded in order to highlight the sensor reliability over time and any changes. According to the authors, the results obtained show that some of the low-cost sensors can be a valid alternative to traditional sensors when the latter have high cost. The case of wind sensors is different, showing significant discrepancy compared to that of the Mesonet network.

I believe that this paper represents a substantial contribution not only in the simple evaluation of low-cost meteorological sensors but also in the study of the robustness of the structure of a 3D printed weather station. In fact, the need of a wider spatial coverage of atmospheric observations at ground (especially in some areas of the Earth) is something that concerns the international scientific community and the deployment of low-cost weather stations (both sensors and structure) can be a suitable and promising answer. However, monitoring weather conditions by low-cost weather stations has its limitations and paper like this can help to quantify these limits.

The paper is interesting, well documented and rather well structured: in my opinion it deserves to be published on AMT as it addresses scientific questions within the scope of the journal. However, I think some changes need to be made before the article is published.

The specific major comments are as follows:

1. A more detailed analysis of the comparisons of the low-cost and reference weather stations is necessary in order to show the clear differences between the two instruments, as the study using only scatterplots and average differences appears too raw and limiting. A more detailed and quantitative approach through merit factors (such as error, bias...) would be desirable.

2. Can be low-cost measurements corrected in some way in order to reproduce reference observations?

3. Are there some meteorological situations/events in which the low-cost station performs best?

4. It is not very clear how the UV data of the two stations were compared, as it stated in the paper that they do not measure the same radiative components.

5. I think a table summarizing all the sensor differences/performances would be valuable to have a clear picture of the comparisons.

6. The comparison between the two rain gauges should be expanded: how the two instruments work on the basis of rain rate?

Minor comments:

1. Do you have any idea about the duration of 3D-printed weather station and its sensors without any maintenance located, for example, in a remote area?

2.Line 64: the average difference of air temperature is 0.81, while the related scatterplot indicates 0.82.
* * *

---

## Referee Comment (RC2) · Anonymous Referee #2 · 16 Mar 2020

Review of the manuscript "More Science with Less: Evaluation of a 3D-Printed Weather Station"

This work presents the results of a comparison between a classical commercial weather station and an innovative 3D-printed system. The experiment lasted eight months and was carried out in Oklahoma (USA), using 5-minutes averaged data of the following meteorological variables: temperature, relative humidity, atmospheric pressure, wind speed and direction and UV data. The authors managed an innovative topic that can be of interest for many readers of AMT journal. The 3D-printed weather station might be an appealing solution for a large number of meteorological applications, in which it is necessary to preserve a balance between instruments price and accuracy. However, from a strictly scientific and technical point of view, the paper has some

relevant point of weaknesses that should be carefully addressed by the authors. As a general comment, the quality presentation of the study is unsatisfactory: therefore, the first suggestion is to perform a formal revision of the manuscript. I think that each of the five sections should be extended and detailed: more information, more discussion and more results are needed. In other words, the manuscript is too short and does not satisfy, in the current version, the minimum standards of an international peer-reviewed scientific journal. Some critical issues and suggestions are provided in the following main comments.

- Introduction: in my opinion, the authors should provide a more detailed and comprehensive state of the art of the considered topic. Moreover, they should better emphasize the added-value of their study compared to the previous work.

- Station configuration: the authors must provide additional details about technical characteristics of each of the meteorological weather stations involved in this study, the commercial one (Mesonet) and the innovative one (3D-printed). More specifically, I suggest adding a table that list the following specifications: range of measure, resolution, update interval, time-constant and uncertainty (or accuracy). Please consider the following WMO manual as reference: World Meteorological Organization: Guide to Meteorological Instruments and Methods of Observation, 2008.

- Deployment: According to Table 2, the traditional weather station includes sensors from different commercial companies (Vaisala, RM Young, Met One, Li-Cor). Why did the authors choose a reference meteorological station with these features and with this configuration? From a comparison with standards required by WMO (see Annex 1.E of WMO, 2008), emerges that those sensors are not an adequate and good benchmark to evaluate the performance of the proposed 3D-printed station. For example, according to WMO recommendations, temperature sensor should have an uncertainty of 0.2 K, which is considerable lower than the uncertainty of the RM Young 41342 RTD Probe (0.5 K). This consideration is easily extendable to other "reference" sensors involved in this study, which do not satisfy the WMO requirements. Probably, the authors chose

the sensors listed in Table 2 as reference because their accuracy is comparable to that of 3D-printed instruments. However, I am quite skeptical about this approach. At first instance, it may be reasonable, but I think that an additional comparison with sensors that fulfill the WMO standard is necessary, in order to achieve results that are valuable from a "high-level" scientific perspective. Moreover, I suggest adding a figure including a photo of Mesonet station facilities. For a reliable comparison, the sensors of the two stations should be installed at the same height above the ground level: as an example, the wind sensors operated at two very different heights (10 m for the Mesonet, 2 m for the 3D-printed station). The authors seem to be conscious of this limit (Lines 85-87), but in my opinion they should discuss this aspect in a more comprehensive manner and should better highlight the limits of their work.

- Results: the measurements of the two meteorological stations have been compared only in terms of simple scatter plots. It is a very "rudimental", although useful, analysis. Therefore, I suggest to do more work in this sense: for example, it may be interesting evaluating the performance of the proposed stations as a function of the season and to investigate about the data accuracy in particular "extreme" weather conditions (e.g. strong winds, cold and/or heat waves, strong rainfall, fog, etc.). Furthermore, for rainfall data, I suggest to perform a comparison not only in terms of daily accumulated rainfall but also in terms of rain rate.

- Conclusions: please add an additional discussion about the limits of those preliminary results and about the future planning and evolution of this study.

Finally, I suggest to carefully checking the paper to address some minor typos.

Best regards.

---

## Author Response (AR1)

**Response to Referee #2**

Thank you for your review of our manuscript. Your comments were well received and will lead to a better paper in response. We will structure our responses as follows: each referee comment will have a number based on referee number and comment from their review. The authors response will be the number with a "R" next to it and changes in the manuscript will be the number with a "C".

**2.1 Introduction: in my opinion, the authors should provide a more detailed and comprehensive state of the art of the considered topic. Moreover, they should better emphasize the added-value of their study compared to the previous work.**

2.1R - Authors agree. There has been limited peer reviewed articles on this topic. C1 This study will provide an argument for the expanded use of these low cost sensors in science and education. The results on the previous studies will be improved with the comparisons against additional commercial instrumentation, but also through improvements to the processing and analysis. Additional corrections to the wind speed, relative humidity, and solar radiation have been performed improving the overall results. This will also provide the first look at the performance of the UV radiation sensor and its ability to indirectly measure the global downwelling solar radiation.

2.1C - The introduction, station configuration, and results sections will be updated accordingly.

**2.2 Station configuration: the authors must provide additional details about technical characteristics of each of the meteorological weather stations involved in this study, the commercial one (Mesonet) and the innovative one (3D-printed). More specifically, I suggest adding a table that list the following specifications: range of measure, resolution, update interval, time-constant and uncertainty (or accuracy). Please consider the following WMO manual as reference: World Meteorological Organization: Guide to Meteorological Instruments and Methods of Observation, 2008.**

2.2R - Authors agree. This information has been compiled for both the 3D-printed sensor and the Mesonet sensors. Furthermore, an additional table was developed to compare the accuracy of the reference instrumentation across the previous comparison of the 3D-PAWS stations.

2.2C - Tables 1 and 2 have been updated to gather this information.

2.3 Deployment: According to Table 2, the traditional weather station includes sensors from different commercial companies (Vaisala, RM Young, Met One, Li-Cor). Why did the authors choose a reference meteorological station with these features and with this configuration? From a comparison with standards required by WMO (see Annex 1.E of WMO, 2008), emerges that those sensors are not an adequate and good benchmark to evaluate the performance of the proposed 3D-printed station. For example, according to WMO recommendations, temperature sensor should have an uncertainty of 0.2 K, which is considerable lower than the uncertainty of the RM Young 41342 RTD Probe (0.5 K). This consideration is easily extendable to other "reference" sensors involved in this study, which do not satisfy the WMO requirements. Probably, the authors chose the sensors listed in Table 2 as reference because their accuracy is comparable to that of 3D-printed instruments. However, I am quite skeptical about this approach. At first instance, it may be reasonable, but I think that an additional comparison with sensors that fulfill the WMO standard is necessary, in order to achieve results that are valuable from a "high-level" scientific perspective.

2.3R - This was a low-cost project and access to sensors that fulfill WMO standards while deployed in the field was not possible. The WMO guide does indicate that operational uncertainty conforming to these requirements will not be met in many instances and are only achievable with the "highest quality sensors and procedures". As such, there are a number of organizations that deploy high quality sensors and implement best practices as they relate to calibration and data quality that can serve as viable reference stations. Oklahoma Mesonet sensors undergo routine maintenance and are rotated out of the field on a regular schedule. Calibrations are performed before and after deployment to the field, leading to well-characterized systems. The Oklahoma Mesonet also has a robust data quality program and the data used from their station has been reviewed and properly quality controlled. These sensors were chosen as they were well-maintained sensors that were also accessible to the research team. The 3D Printed weather station was donated to the Cooperative Institute of Mesoscale Meteorological Studies (CIMMS) Education and Outreach program and an additional study will not be possible without building a completely new system. Three of the four authors have also relocated to new positions and funding would need to be secured for another comparison study with sensors meeting WMO standards while deployed in the field.

2.3C - Additional discussion on why these sensors were chosen and how they compare with other studies and the WMO guidelines will be added into the manuscript.

2.4 Moreover, I suggest adding a figure including a photo of Mesonet station facilities. For a reliable comparison, the sensors of the two stations should be installed at the same height above the ground level: as an example, the wind sensors operated at two very different heights (10 m for the Mesonet, 2 m for the 3D-printed station). The authors seem to be conscious of this limit (Lines 85-87), but in my opinion they should discuss this aspect in a more comprehensive manner and should better highlight the limits of their work.

2.4R - Authors agree on adding an additional plot of the Mesonet station. With respect to the wind speed comparisons, a logarithmic wind profile correction was applied to the Mesonet wind speed based on Allen et al 1998. This did bring the Mesonet wind speed values more in line with the 3D printed station for a majority of the deployment.

2.4C - Will add an addition plot of the Mesonet Facility. Authors will also further discuss the differences in the systems and the limits of the comparison.

2.5 Results: the measurements of the two meteorological stations have been compared only in terms of simple scatter plots. It is a very "rudimental", although useful, analysis. Therefore, I suggest to do more work in this sense: for example, it may be interesting evaluating the performance of the proposed stations as a function of the season and to investigate about the data accuracy in particular "extreme" weather conditions (e.g. strong winds, cold and/or heat waves, strong rainfall, fog, etc.).

2.5R - A more in-depth analysis has been done to calculate standard error of means, root mean square error (RMSE), and also a linear regression including slope, intercept, and correlation. This information along with min/max values for each station are included on the plots. This information was also calculated for each month of the deployment and the RMSE and correlation coefficients were recorded in an additional table. Additional processing has been applied to the relative humidity, solar radiation and wind speeds as well. The relative humidity was corrected using a temperature coefficient correction supplied by the vendor. The solar radiation was measured in counts and a linear regression was performed to convert counts to W/m2. Given the differences in height of wind measurement heights, a logarithmic wind profile conversion was done based on Allen et al 1998. The station was only deployed for 8 months, so more data would be needed to evaluate the performance of the station as a function of season and to further investigate data accuracy in extreme weather conditions. Interestingly, the precipitation gauge performed very well in the one heavier rain event that was encountered. The 3D printed gauge and Mesonet both recorded roughly 104 mm/hr precipitation rate.

2.5C - Authors will update figures with more in depth statistics, summary table of RMSE and correlation coefficient, and additional discussion on results.

2.6 Furthermore, for rainfall data, I suggest to perform a comparison not only in terms of daily accumulated rainfall but also in terms of rain rate.

2.6R- Data were processed for rain rate and will be included in the discussion and results accordingly.

2.6C - Rain rate results will be added to the results section.

2.7 Conclusions: please add an additional discussion about the limits of those preliminary results and about the future planning and evolution of this study.

2.7R - Authors will add additional discussion noting the differences between the systems and the limitations. The future and evolution of this study would have naturally been to extend it to different measurement types (soil moisture, spectral radiation, etc). Two of the authors have/will graduate from the University and the lead author has changed jobs and is now at Argonne National Lab in Chicago which will limit additional comparisons with the Oklahoma Mesonet. However, there still exists some opportunities to further advance this topic and potentially partner with other groups for future studies.

2.7C - Authors will add in addition discussions on the limits of the results and the natural evolution of this study

**Response to Referee #3**

Thank you for your review of our manuscript. Your comments were well received and will lead to a better paper in response. We will structure our responses as follows: each referee comment will have a number based on referee number and comment from their review. The authors response will be the number with a "R" next to it and changes in the manuscript will be the number with a "C".

**3.1 - A more detailed analysis of the comparisons of the low-cost and reference weather stations is necessary in order to show the clear differences between the two instruments, as the study using only scatterplots and average differences appears too raw and limiting. A more detailed and quantitative approach through merit factors (such as error, bias...) would be desirable.**

3.1R - Authors agree. Additional processing has been applied to relative humidity, solar radiation and wind speeds. A correction for the temperature coefficient was applied to the relative humidity based on vendor documentation which improved the overall results. The solar radiation was measured in counts and a linear regression was performed to convert counts to W/m2. Given the differences in height of wind measurement heights, a logarithmic wind profile conversion was done based on Allen et al 1998. Using these results, a more in-depth analysis has been performed to calculate standard error of means, root mean square error (RMSE), and also a linear regression including slope, intercept, and correlation. This information along with min/max values for each station are included on the plots. Statistics were also calculated for each month of the deployment and the RMSE and correlation coefficients were recorded in an additional table. Additionally, the performance of the temperature and relative humidity sensors were analyzed with increasing wind speeds to determine the effects the naturally aspirated wind shield would have on the measurements.

3.1C - Updated figures with more in depth statistics, summary table of RMSE and correlation coefficient, summary table of RMSE and correlation coefficient for temperature and relative humidity based on different wind speed thresholds, and additional discussion on results.

**3.2. Can be low-cost measurements corrected in some way in order to reproduce reference observations?**

3.2R - It would be relevant to run the sensors through calibrations before and after deployment to the field to better characterize the sensors, similar to the Oklahoma Mesonet practice. However this would also add additional expenses. Inaccuracies owing to drift would be harder to compensate for. It is also unknown how the drift rates differ between boards of the same sensor type. Further analysis of the performance of multiple boards of the same sensor would be necessary to better characterize them. As mentioned in 3.1R, the relative humidity, solar radiation, and wind speed measurements were corrected, which produced improved results.

3.2C - Where relevant, the result discussion was updated with the corrected data.

**3.3. Are there some meteorological situations/events in which the low-cost station performs best?**

3.3R - Through the added analysis, it was determined that the temperature sensors had improved performance with increasing wind speed, which is expected as the flow through the radiation shield would be more comparable to the Mesonet aspirated radiation shields. The relative humidity did not see the same improvement in performance which could be related to dust collection on the filter on the sensor. While the filter on the sensor was hydrophobic, the dust that collected on it may have not been. The relative humidity did perform better in drier conditions which is expected based on the reduce vendor stated accuracy for the lower to mid ranges of relative humidity. Overall though, more data over repeated seasons would be necessary to determine which conditions/events the station performs best in.

3.3C - Manuscript results discussion was updated with some of this additional analysis.

**3.4. It is not very clear how the UV data of the two stations were compared, as it stated in the paper that they do not measure the same radiative components.**

3.4R - The UV sensor calculates the UV index by measuring visible and infrared light. This is stored as counts in the data files. It was determined that some other sensors did provide equations to calculate lux from which the W/m2 could be estimated, the sensor used in this study did not. Authors initially used the visible counts to compare with the downwelling global solar radiation but the errors reported would not be informative. To improve on the analysis, a linear regression was performed on the entire dataset and a slope/intercept was applied to the data to make the measurements comparable.

3.4C - Manuscript discussion on the radiation results, solar radiation plot, and statistics will be updated with the new results.

**3.5. I think a table summarizing all the sensor differences/performances would be valuable to have a clear picture of the comparisons.**

3.5R - Authors agree.

3.5C - Table 4 will be included with summary of RMSE and correlation coefficient for all measurements for each month and for the entire campaign. This is also relevant to updates made in Table 1 and Table 2 to better follow WMO guidelines on reporting range, resolution, accuracy, and more. Table 3 was added to give an overview of the specific measurement accuracy of the reference sensors used thus far in all comparisons with 3D-PAWS.

**3.6. The comparison between the two rain gauges should be expanded: how the two instruments work on the basis of rain rate?**

3.6R - Both rain gauges are tipping bucket gauges with the 3D printed rain gauge performing a tip for every 0.2 mm while the Mesonet bucket tips on 0.254 mm. The white Mesonet gauge is

located 0.6 m off the ground and is surrounded by an alter shield to decrease the wind effects. The gray 3D weather station gauge was roughly 0.3 m off the ground and was not surrounded by an alter shield. The color of the gauges are noted as neither gauge has a heater and how the different gauges heat up following a solid precipitation event could impact the rain rates recorded.

3.6C - Manuscript will be updated to include this relevant information and expand on it. Minor Comments

3.m1 - Do you have any idea about the duration of 3D-printed weather station and its sensors without any maintenance located, for example, in a remote area?

3.m1R - Based on our experience, the durability of the 3D printed parts varied. Printed parts used in the wiring of the station such as the nuts for tightening instruments in place, for example, would have a low life expectancy without proper maintenance. The durability of the wiring connectors was inconsistent due to minor imperfections caused by the printing process. These imperfections forced excessive filing of the wire connectors in order to securely fit wiring. Connectors that were excessively filled often wore down to the point where wiring had difficulty remaining in the connector itself. Nuts tended to lose grip over time causing the instruments to sag or rotate in their housing. Parts with constant friction, like the anemometer, would have lower expected life spans. Most of the parts that did not fall under these previous examples remained in very good condition at the conclusion of the deployment. Sensor longevity seemingly was related to whether the sensor could be impacted by the outside elements, namely moisture. The temperature, humidity, and pressure sensors housed in the radiation shield showed varying degrees of corrosion throughout the study, with the humidity sensor ultimately having to be removed. The rain gauge and wind sensors, which were more sheltered from the outside conditions, showed no evidence of degradation at the conclusion of the study. If we were to recreate this system again, the wiring connectors would be the main area of improvement sought as a lot of time was spent working on these connections. The fitting to secure the wind direction vane would be second as the results were not ideal as it kept coming loose and rotating from truth North orientation.

3.m1C - Manuscript section discussing the 3D printed components will be updated with some additional information.

3.m2 Line 64: the average difference of air temperature is 0.81, while the related scatterplot indicates 0.82

2.m2R - Thank you for noting this. The more in-depth analysis has produced some updated results and the manuscript will be updated accordingly.

**Changes to Manuscript**

Based on the referee comments, major revisions were made to the manuscript. This included updates to all sections. An overview of the updates are noted below, followed by the full paper showing differences as detected by latexdiff.

- Introduction
  - Major additions to better scope the relevance of this work along with introducing the reference instrumentation a little more.
- Station Configuration
  - The deployment section was split out and added to introduction and station configuration
  - Additional information was added to the station configuration section to denote the differences in the 3D-printed system vs the Oklahoma Mesonet.
- Results
  - Results were completely rewritten to include additional analysis that was request, including RMSE, Correlation Coefficient, and more.
- Discussion
  - Discussion was updated based on new and improved results
- Appendix A
  - Added to better document the instrumentation used in previous comparisons using the 3D-PAWS system
- Figures
  - All graphs where updated with new results
  - Image of the Norman Mesonet site was added
- Tables
  - Table 1 was updated to include instrument specifications for the 3D-PAWS sensors
  - Table 2 was updated to include additional instrument specifications from the Mesonet station
  - Table 3 was added to show the reference instrument accuracies from past 3D-PAWS studies
  - Table 4 was added to reference RMSE and correlation coefficient for each month of the deployment and then overall values
  - Table 5 was added to document the effects of wind speed on the temperature and humidity results
  - Table 6 was added to document the instrument performance as it relates to rain accumulation and rain rate through the deployment.

[revised manuscript text omitted]

Wind speed resolution from Kucera and Steinson results

Mode definitions: I = Instantaneous measurement; A = Average over time; T = Total

**Table 2.** Oklahoma Mesonet instrumentation (Mcpherson et al., 2007)

| Parameter | Sensor |  Range | Resolution | Acc... |
|---|---|---|---|---|
| Air Temperature | RM Young 41342 RTD Probe | -50–50 °C (calibrated) | 0.01 °C  |  |
| Relative  Humidity | Vaisala  HMP155 |  0–100 % | 0.03 % | ± 1 % (... ± 0.6 %... |
| Atmospheric  Pressure | Vaisala  Barometer | 500–1100 hPa | 0.1 hPa | ±  ... |
| Wind Speed  | RM Young  Wind Monitor | 0–100 m s$^{-1}$ | 0.03 m s$^{-1}$ | ± 1 % or ... |
| Wind  Direction | RM Young  Wind Monitor | 0–360 ° | 0.05 ° | ± 3... |
| Solar  Radiation | Li-Cor Pyranometer | 0–3000 W m$^{-2}$ | 0.23 W m$^{-2}$ | ± ... |
| Rainfall  | Met One TBRG | | 0.25 mm | 1 % (2.5–7... at 2... |

Mode definitions: I = Instantaneous measurement; A = Average over time; T = Total

Temperature accuracy does not include the added uncertainty from the radiation shield

**Table 3.** Summary of reference instrument accuracy used in 3D-PAWS comparison studies, including requirements from WMO Guide to Instrument and Methods of Observation - Volume 1 Annex 1.A. (WMO, 2018)

| Parameter | Smallwood & Santarsiero | Aura et al. | Kucera & Steinson NCAR Testbed | Kucera & Steinson NOAA Testbed | Oklahoma Mesonet | WMO Guidelines* |
|---|---|---|---|---|---|---|
| Air Temperature | 1.1 °C | 0.6 °C | 0.1 °C at 23 °C | 0.28 °C (-50–50 °C) | 0.3 °C at 23 °C | 0.1 °C (-40–40 °C) AMU: 0.2 °C |
| Relative Humidity | 5 % (90–100 %) 4 % (80–90 %) 3 % (20–80 %) 4 % (10–20 %) 5 % (1–10 %) | 4 % (90–100 %) 2 % (15–90 %) | 0.8 % at 23 °C | Dewpoint Temperature 1 °C (-1–30 °C) | ± 1 % (40–97 %) ± 0.6 % (0–40 %) | 1 % AMU: 3 % |
| Atmospheric Pressure | | 1 hPa | 0.5 hPa | 0.1 hPa | 0.2 hPa | 0.1 hPa AMU: 0.15 hPa |
| Wind Speed | Accuracy in m s$^{-1}$ 2.2 (<44 m s$^{-1}$) 1.8 (<22 m s$^{-1}$) 1.3 (<13 m s$^{-1}$) 0.9 (<4.5 m s$^{-1}$) | 3 % | Greater of 0.3 m s$^{-1}$ or 3 % | Greater of 0.135 m s$^{-1}$ or 3 % | Greater of 1 % or 0.3 m s$^{-1}$ | 0.5 m s$^{-1}$ (<5 m s$^{-1}$) 10 % (> 5 m s$^{-1}$) AMU: Not Listed |
| Wind Direction | | 5 ° | 3 ° | 2 ° | 3 ° | 5 ° AMU: 5 ° |
| Solar Radiation | NA | 5 % | NA | NA | 5 % | 2 % AMU: Daily: 5 % AMU: Hourly 8 % |
| Rainfall | 5 % | 5 % | 0.1 % FS | 4 % | 1 % (2.5–7.6 cm hr$^{-1}$) at 21 °C | 0.1 mm (<5mm) 2 % (> 5mm) AMU: Greater of 5 % or 0.1 mm |

AMU: WMO Achievable Measurement Uncertainty

* - WMO Guide to Instruments and Methods of Observation, Volume 1 - Measurement of Meteorological Variables, Annex 1.A (WMO, 2018)

Information was retrieved from a number of sources, see Appendix A for details

Temperature accuracy does not include the added uncertainty from the radiation shield

**Table 4.** Comparison statistics summary of RMSE (top value) and correlation coefficient (bottom value)

| Parameter | Month 1 15 Aug 2018 | Month 2 16 Sep 2018 | Month 3 16 Oct 2018 | Month 4 16 Nov 2018 | Month 5 16 Dec 2018 | Month 6 16 Jan 2019 | Month 7 16 Feb 2019 | Month 8 16 Mar 2019 | Entire Period |
|---|---|---|---|---|---|---|---|---|---|
| MCP Air Temperature (°C) | 1.32 0.98 | 1.06 0.99 | 1.26 0.99 | 1.33 0.99 | 1.20 0.99 | 0.90 0.99 | 0.42 1.00 | 1.53 0.98 | 1.22 1.00 |
| HTU Air Temperature (°C) | 1.11 0.99 | 0.94 0.99 | 0.96 1.00 | 0.99 0.99 | 0.85 0.99 | 1.00 0.99 | | | 0.97 1.00 |
| Relative Humidity (%) | 2.63 0.99 | 3.38 0.99 | 3.73 0.99 | 3.42 0.99 | 3.45 0.99 | 3.09 0.99 | | | 3.33 0.99 |
| Atmospheric Pressure (hPa) | | 2.92 1.00 | 2.51 1.00 | 2.21 1.00 | 2.44 1.00 | 2.60 1.00 | 2.41 1.00 | 1.90 1.00 | 2.39 1.00 |
| Wind Speed (m s$^{-1}$) | 0.58 0.92 | 0.56 0.94 | 0.59 0.95 | 0.67 0.95 | 0.69 0.94 | 0.88 0.94 | 2.49 0.67 | 2.98 0.29 | 1.47 0.74 |
| Wind Direction (°) | 79.74 0.60 | 68.03 0.75 | 80.13 0.80 | 95.02 0.74 | 87.16 0.78 | 118.58 0.42 | 94.49 0.73 | 130.27 0.63 | 96.52 0.68 |
| Solar Radiation (W m$^{-2}$) | 81.64 0.98 | 68.77 0.96 | 50.86 0.98 | 52.35 0.99 | 44.49 0.98 | 34.14 0.99 | 34.23 0.99 | 54.22 0.99 | 58.05 0.97 |
| Rainfall Daily Total (mm) | 0.65 1.00 | 1.32 1.00 | 0.27 1.00 | 2.21 0.98 | 3.79 0.92 | 0.26 0.89 | 9.87 0.98 | 8.02 0.25 | 4.68 0.87 |
| Rain Rate (mm hr$^{-1}$) | 0.42 0.96 | 0.61 0.94 | 0.35 0.87 | 0.34 0.86 | 0.69 0.91 | 0.20 0.33 | 1.27 0.88 | 1.25 0.01 | 0.74 0.82 |

Date indicated in first row is the start date of the period used in the analysis.

**Table 5.** 3D weather station temperature RMSE (top value) and correlation coefficient (bottom value) response to increased wind speeds thresholds

| Sensor | $0\ \mathrm{m\,s}^{-1}$ | $1\ \mathrm{m\,s}^{-1}$ | $2\ \mathrm{m\,s}^{-1}$ | $3\ \mathrm{m\,s}^{-1}$ | $4\ \mathrm{m\,s}^{-1}$ | $5\ \mathrm{m\,s}^{-1}$ | $6\ \mathrm{m\,s}^{-1}$ | $7\ \mathrm{m\,s}^{-1}$ | $8\ \mathrm{m\,s}^{-1}$ |
|---|---|---|---|---|---|---|---|---|---|
| MCP9808 Temperature (°C) | 1.22 1.00 | 1.08 1.00 | 1.01 1.00 | 0.94 1.00 | 0.88 1.00 | 0.75 1.00 | 0.60 1.00 | 0.54 1.00 | 0.51 1.00 |
| HTU21D Temperature (°C) | 0.97 1.00 | 0.91 1.00 | 0.87 1.00 | 0.83 1.00 | 0.77 1.00 | 0.65 1.00 | 0.50 1.00 | 0.43 1.00 | 0.47 1.00 |
| HTU21D Relative Humidity (%) | 3.33 0.99 | 3.35 0.99 | 3.28 0.99 | 3.27 1.00 | 3.33 1.00 | 3.41 1.00 | 3.56 1.00 | 3.59 1.00 | 3.49 1.00 |

**Table 6.** Maximum precipitation rate and daily accumulations recorded each month

| Parameter | Month 1 15 Aug 2018 | Month 2 16 Sep 2018 | Month 3 16 Oct 2018 | Month 4 16 Nov 2018 | Month 5 16 Dec 2018 | Month 6 16 Jan 2019 | Month 7 16 Feb 2019 | Month 8 16 Mar 2019 |
|---|---|---|---|---|---|---|---|---|
| Mesonet Accumulation (mm) | 53.1 | 66.6 | 22.6 | 15.8 | 37.1 | 1.8 | 26.9 | 37.9 |
| 3D-Printed Accumulation (mm) | 52.8 | 72.0 | 22.4 | 26.6 | 42.6 | 2.2 | 75.2 | 7.2 |
| Mesonet Rain Rate (mm hr$^{-1}$) | 42.7 | 51.8 | 27.5 | 42.6 | 103.7 | 6.1 | 36.6 | 36.6 |
| 3D-Printed Rain Rate (mm hr$^{-1}$) | 43.2 | 50.4 | 26.4 | 33.6 | 103.2 | 7.2 | 86.4 | 57.6 |

---

## Referee Report (RR1)

The authors have thoroughly addressed concerns and remarks from the reviewers. As a consequence, the paper has been modified and improved considerably.

However, I believe that some other corrections and/or clarifications are needed before publication.

The specific comments are as follows:

1. The exact period and dates of the comparison between the two AWS should be emphasized in the text. It is only reported in the abstract, while, in line 102, it is worth wondering whether these dates correspond only to the scale of the scatterplots or are instead the actual period of comparison.
2. Clarifications appear necessary about the general evaluation of the 3D-Printed WS. Also, the shortcoming of the durability of the low-cost sensors, as well as of the 3D-Printed WS, should be highlighted.
   I mean: the comparison lasted eight months. Temperature sensor began to fail in month seven; RH sensor in month six; Atmospheric pressure sensor was replaced at the beginning and then suffered some communication problems. Wind speed sensor worked well until month 7, whereas wind vane was problematic for the entire period. Solar radiation sensor suffered from different bias (before high and then low). Rain gauge tended towards high bias starting from month seven. Because of this, some conclusions appear to be pretty daring. Some of the low-cost sensors worked very well but for a limited period of the comparison. Furthermore, 3D printed parts gave some concerns in terms of durability. Hence, the limited time in which these low-cost sensors can serve as viable alternatives to commercial weather stations should be pointed out.
3. Line 193: You mean the maximum rain rate of the heavy precipitation event, I guess. If so, it should be added.

Typos:

Line 47 "are"

Line 79: "Raspberry" for consistency

Line 127: "are"

Line 161: "extend"

Line 216: "is"

Line 229: "low-cost" for consistency

Line 233: "is"

---

## Author Response (AR2)

First off, thank you again to both referees for reviewing these updates.  The itemized response is listed below with the referee comment bolded, followed by the author's response and noted manuscript changes.

**Response to Referee #3 (Report #1)**

**1. The exact period and dates of the comparison between the two AWS should be emphasized in the text. It is only reported in the abstract, while, in line 102, it is worth wondering whether these dates correspond only to the scale of the scatterplots or are instead the actual period of comparison.**

[Author's Response] – The period used for the data comparison and analysis was the same as the deployment of the station, with the exception of the problematic sensors as noted in the text.

[Manuscript Changes] – The specific dates of the deployment were added into line 91 and it was noted in line 104 that the data analysis corresponds to the full length of the deployment.

**2. Clarifications appear necessary about the general evaluation of the 3D-Printed WS. Also, the shortcoming of the durability of the low-cost sensors, as well as of the 3D-Printed WS, should be highlighted.  I mean: the comparison lasted eight months. Temperature sensor began to fail in month seven; RH sensor in month six; Atmospheric pressure sensor was replaced at the beginning and then suffered some communication problems. Wind speed sensor worked well until month 7, whereas wind vane was problematic for the entire period. Solar radiation sensor suffered from different bias (before high and then low). Rain gauge tended towards high bias starting from month seven.  Because of this, some conclusions appear to be pretty daring. Some of the low-cost sensors worked very well but for a limited period of the comparison. Furthermore, 3D printed parts gave some concerns in terms of durability. Hence, the limited time in which these low-cost sensors can serve as viable alternatives to commercial weather stations should be pointed out.**

[Author's Response] – Valid argument.  Agree that the authors should indicate time frames during which these could serve as viable alternatives.  It is worth noting that long-term operations would be feasible if routine maintenance and replacement of sensors and 3D-printed components were performed on a regular basis (4-6 months).

[Manuscript Changes] – Abstract and conclusions (lines 240-245) have been reworked to note that these sensors could serve as viable alternatives for short-term deployments and longer-term deployments would need routine replacement of components.

**3. Line 193: You mean the maximum rain rate of the heavy precipitation event, I guess. If so, it should be added.**

[Author's Response] – Yes, it was the maximum rain rate of the precipitation event.

[Manuscript Changes] – Line 203 was updated to denote it was the maximum precipitation rate

**Typos**
Typos have been addressed and we thank the referee for pointing those out.

**Response to Referee #2 (Report #2)**

**They performed a good work, although the analysis section still remain confined to a basic level. For example, no additional focus on a specific relevant meteorological event has been performed. The authors cited only a heavy rainfall event (in which a rain rate over 100 mm/h has been detected by both sensors), but they did not present a detailed investigation producing new figures and/or new discussions.**

[Author's Response] – The goal of this study was to assess the long-term performance of these sensors and 3D-printed components.  Although the deployment was cut short from our year-long goal, we do believe we've captured enough information to analyze the performance of the sensors and 3D-printed components through the "bulk" analysis that was performed.  The authors reviewed possible case studies that could be utilized in the analysis, such as the squall line event that produce the maximum precipitation rate recorded in month five, however, the insights that could be gained from these limited periods were already captured in the existing discussion.  While response times between these systems could have been a focus, higher temporal resolution data would be needed from the Mesonet station.  Instead of a case study, we chose to refine the statistical analysis, discuss relevant corrections that were implemented to improve the results, including easily overlooked vendor corrections, and further investigate the overall differences between the systems.

[Manuscript Changes] – No changes

**Considering the novelty of the topic, I suggest to accept the paper after a minor revision. The latter involved mainly the Figure 10, which is difficult to read. In my opinion, the authors should produce a wind rose plot which may be a good solution to synthetize the difference in performance between the 3D-Station and the reference Mesonet sensor.**

[Author's Response] – Agreed.  The graph is noisy with the degraded performance of the wind vane.

[Manuscript Changes] – Figure 10 has been replaced with a wind rose plot and the wind direction results discussion has been updated.

**Moreover, in the caption of Figures 5-13, the authors must clarify the difference between the**

**dashed blue line and the solid black line. I have some doubts about the realiability of the linear regression model of the Figure 9. Please verify the suitability of this linear fit to the data. Moreover, in fig. 9 I see many points placed along y-axis, meaning that calm wind (0.0 m/s) has been measured by 3D-Station; I suppose that those points represent cases of 3D-sensor fault. I suggest to eliminate these cases from the scatter plot and to reproduce, for this figure and also for the other ones, only the cases in which the two sensors worked properly.**

[Author's Response] – The anemometer started suffering from problems on 16 February 2018 with intermittent dropouts with a full failure on 30 March 2019.  Removing these points from the analysis does improve the linear fit, improving the correlation coefficient from 0.74 to 0.86 m s$^{-1}$.  It can also be noted that the data from February on does skew the results significantly.  In Table 4, the wind speed RMSE and correlation coefficient during the first six months (RMSE < 0.9, CC > 0.9) is significantly better than the last 2 months (RMSE >1, CC < 0.9).  Additionally, wind speeds of 0 m s$^{-1}$ have been removed from the wind rose plot after the 16 February 2018 date.

As for the differences in the lines, these are noted in the results section, but agree it would be beneficial to include in the plot captions as well.

[Manuscript Changes] – Results section for wind speed and direction have been updated with the new results and discussion from exclude the problematic data.  Fig 9 was reproduced with data excluded from analysis.  Table 4 was also updated with new RMSE and correlation coefficient calculations. Scatter plots captions have been updated to describe the blue and black lines.

**List of Manuscript Changes**

[Manuscript Changes] – The specific dates of the deployment were added into line 91 and it was noted in line 104 that the data analysis corresponds to the full length of the deployment.

[Manuscript Changes] – Abstract and conclusions (lines 240-245) have been reworked to note that these sensors could serve as viable alternatives for short-term deployments and longer-term deployments would need routine replacement of components.

[Manuscript Changes] – Line 203 was updated to denote it was the maximum precipitation rate

[Manuscript Changes] – Figure 10 has been replaced with a wind rose plot and the wind direction results discussion has been updated.

[Manuscript Changes] – Results section for wind speed and direction have been updated with the new results and discussion from exclude the problematic data.  Fig 9 was reproduced with data excluded from analysis.  Table 4 was also updated with new RMSE and correlation coefficient calculations. Scatter plots captions have been updated to describe the blue and black lines.

[revised manuscript text omitted]